# MEMORY-T1: REINFORCEMENT LEARNING FOR TEMPORAL REASONING IN MULTI-SESSION AGENTS

**Yiming Du**[1,2]**, Baojun Wang**[3]**, Yifan Xiang**[1,2]**, Zhaowei Wang**[4]**, Wenyu Huang**[5]**,
Boyang Xue**[1,2]**, Bin Liang**[1,2]**, Xingshan Zeng**[3]**, Fei Mi**[3]**, Haoli Bai**[3]**, Lifeng Shang**[3]**,
Jeff Z. Pan**[5]**, Yuxin Jiang**[3]*,**Kam-Fai Wong**[1,2]*

[1]The Chinese University of Hong Kong
[2] MoE Key Laboratory of High Confidence Software Technologies, China
[3]Huawei Technologies Co.,Ltd [4]HKUST [5]The University of Edinburgh
{ydu, kfwong}@se.cuhk.edu.hk, jiang.yuxin2@huawei.com

## ABSTRACT

Temporal reasoning over long, multi-session dialogues is a critical capability for conversational agents. However, existing works and our pilot study have shown that as dialogue histories grow in length and accumulate noise, current long-context models struggle to accurately identify temporally pertinent information, significantly impairing reasoning performance. To address this, we introduce **MEMORY-T1**, a framework that learns a time-aware memory selection policy using reinforcement learning (RL). It employs a coarse-to-fine strategy, first pruning the dialogue history into a candidate set using temporal and relevance filters, followed by an RL agent that selects the precise evidence sessions. The RL training is guided by a multi-level reward function optimizing (i) **answer accuracy**, (ii) **evidence grounding**, and (iii) **temporal consistency**. In particular, the temporal consistency reward provides a dense signal by evaluating alignment with the query time scope at both the session-level (chronological proximity) and the utterance-level (chronological fidelity), enabling the agent to resolve subtle chronological ambiguities. On the Time-Dialog benchmark, Memory-T1 boosts a 7B model to an overall score of 67.0%, establishing a new state-of-the-art performance for open-source models and outperforming a 14B baseline by 10.2%. Ablation studies show temporal consistency and evidence that grounding rewards jointly contribute to a 15.0% performance gain. Moreover, Memory-T1 maintains robustness up to 128k tokens, where baseline models collapse, proving effectiveness against noise in extensive dialogue histories. The code and datasets are publicly available at https://github.com/Elvin-Yiming-Du/Memory-T1/

## 1 INTRODUCTION

Recent advances in memory architectures and large language models (LLMs) have substantially improved the capabilities of conversational agents (Yu et al., 2025; Zhong et al., 2024; Xu et al., 2025). Increasingly, these agents are expected to support long-term multi-session interactions (Du et al., 2025b; Ge et al., 2025), where a central challenge is understanding and reasoning about temporal relationships across dialogue histories (Wu et al., 2025; Maharana et al., 2024). Without this capability, agents may incorrectly order past events, conflate information from different sessions, and ultimately generate inconsistent or inaccurate answers. For example, as shown in Figure 1, correctly resolving a query such as "What time did Emi mention that some 'Suits' characters were together at the Golden Globes?" requires the agent to locate the relevant mention in the dialogue history, understand the key relative temporal expression ("last night"), and grounding it to the correct session date ("10.01.2024") to infer the accurate date "January 9, 2024". Ultimately, temporal reasoning is essential for factual consistency in long, noisy conversations.

However, existing approaches (Yu et al., 2025; Xu et al., 2025) remain inadequate for temporal reasoning in conversation. General-purpose long-context models (Team, 2024; Guo et al., 2025b;

---

*Corresponding authors.

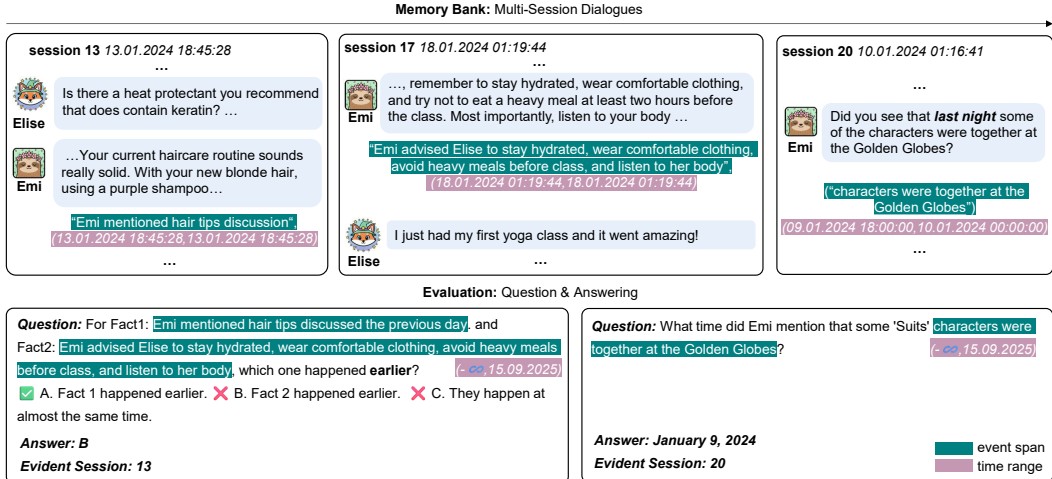

Figure 1: Multi-session QA with time-event annotations. Time range marks when an event or query occurs, either a duration or an instantaneous point (start and end coincide). Event span highlights key evidence in the utterance.

Wang et al., 2025) treat dialogue history as flat text and fail to locate or resolve temporal expressions, leading to steep performance degradation on noisy, extensive conversations (Wu et al., 2025; Maharana et al., 2024). Time-aware frameworks such as TReMu (Ge et al., 2025) handle explicit expressions but struggle with ambiguous ones like "the week before that", and error accumulation from inferred event summaries undermines robustness. Reinforcement learning (RL) approaches such as Time-R1 (Liu et al., 2025) rely heavily on structured metadata, making them ineffective for unstructured multi-session dialogues. Thus, a robust, scalable solution for temporal reasoning in dialogue remains an open challenge.

To bridge this gap, we introduce **Memory-T1**, a RL-based memory retrieval framework designed for temporal reasoning that combines coarse-to-fine retrieval strategy with a multi-level reward design. In the **Candidate Generation** phase, the query temporal scope is predicted using an LLM to prune the dialogue history search space, which acts as a hard filter to prune irrelevant sessions. This is followed by a relevance-based retriever to produce a small, high-recall candidate set of sessions. This phase efficiently narrows the vast memory pool to a manageable context, setting the stage for a more precise analysis. In the **fine-grained selection** phase, an RL agent identifies the precise evidence sessions. Training such an agent is challenging because answer-only supervision provides very sparse signals. To overcome this, we design a dense, multi-level reward function. Beyond answer accuracy ($R_a$) and evidence grounding ($R_g$), we introduce a novel temporal consistency reward ($R_t$) that explicitly evaluates (1) session-level chronological proximity and (2) utterance-level temporal density. By rewarding temporally coherent and contextually concentrated evidence, this structured signal provides richer supervision, enabling the agent to resolve ambiguous temporal expressions and to generalize more robustly to noisy, long-context dialogues.

We validate Memory-T1 on the Time-Dialog (Wei et al., 2025) and LoCoMo (Maharana et al., 2024) benchmarks. Results show that Memory-T1 achieves state-of-the-art temporal reasoning performance, substantially improving robustness on contexts up to 128k tokens. Notably, Memory-T1 enables a 7B model to outperform a 14B baseline, highlighting the effectiveness of temporal-aware retrieval and dense reward optimization. The key contributions are: (1) A coarse-to-fine memory retrieval framework that efficiently narrows dialogue histories into high-quality candidates before fine-grained evidence selection. (2) A novel dense reward design for RL-based retrieval introducing temporal consistency signals at both session and utterance levels, providing insights into training robust temporal-aware retrieval models by overcoming sparse reward limitations. (3) State-of-the-art performance, with Memory-T1 achieving top results and maintaining accuracy under extremely long and noisy conversational contexts.

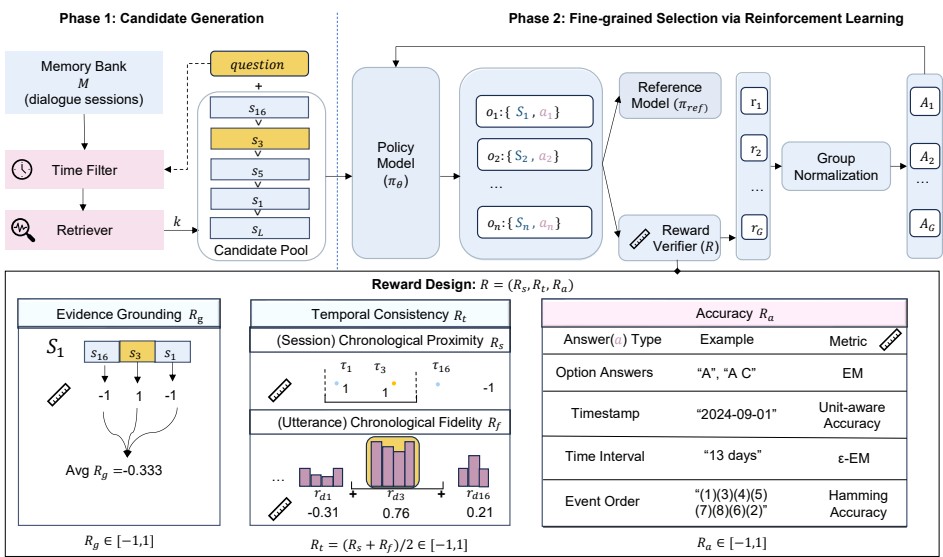

Figure 2: An overview of Memory-T1. The framework employs a coarse-to-fine cascade to select time-consistent memories for multi-session temporal reasoning.

## 2 RELATED WORK

**Temporal Reasoning in LLMs.** Temporal reasoning has become an active area of research for LLMs (Song et al., 2025; Wei et al., 2025; Liu et al., 2025). Benchmarks such as TimeBench (Chu et al., 2024) and TIME (Wei et al., 2025) reveal that even strong models struggle with temporal relationships, event ordering, factual consistency, and long-range reasoning. To fill these gaps, prior work has aligned knowledge with temporal contexts (Zhao et al., 2024), introduced specialized training such as Timo (Su et al., 2024) and TG-LLM (Xiong et al., 2024), or applied RL, as in DeepSeek-R1 (Guo et al., 2025b) and Time-R1 (Liu et al., 2025). However, these methods often depend on explicit supervision or handcrafted structures, limiting their applicability to multi-session dialogue. Furthermore, temporal reasoning has recently become an important problem in the memory of LLMs. TReMu (Ge et al., 2025) leverages memory via timeline summaries but relies on timestamp accuracy for temporal reasoning. A few memory-related works (Mai et al., 2025; Du et al., 2025a) also highlight the importance of temporal reasoning in long-term memory modeling. Building on this perspective, our Memory-T1 framework directly learns implicit memory selection and temporal alignment through a multi-level time consistency reward, enabling robust reasoning without external tools.

**Reinforcement Learning in Agents:** Reinforcement learning is a core technology driving breakthroughs in LLM reasoning, from early outcome-based optimization algorithms, such as PPO (Schulman et al., 2017), to recent variants for agent scenarios, such as GRPO (Zheng et al., 2025), DPO (Rafailov et al., 2023), and GSPO (Zheng et al., 2025). RL not only improves training stability but also efficiency, enabling reasoning-centric models like DeepSeek-R1 (Guo et al., 2025a) and Search-R1 (Jin et al., 2025). Beyond isolated reasoning, RL has been applied to agent settings involving tool use (Qian et al., 2025), multi-step planning (Jin et al., 2025), and long-term interaction (Yu et al., 2025). Recent studies further extend RL to diverse scenarios, including optimized tool integration (Li et al., 2025), emergent code execution under large-scale training (Mai et al., 2025), and generalized frameworks for retrieval and collaboration (Luo et al., 2025). However, temporal reasoning over multi-session dialogues remains an underexplored area, necessitating robust memory retrieval, chronological alignment of events, and reasoning with ambiguous supervision.

## 3 MEMORY-T1

Temporal reasoning over extended, multi-session dialogues presents a significant challenge in conversational AI. The task requires agents to navigate vast and noisy memory banks to retrieve tem-

porally accurate and contextually relevant information, a process where existing models often fail. To address this, we propose Memory-T1, a novel framework for temporal-aware memory retrieval. We proceed as follows: Section 3.1 provides a formal problem definition, Section 3.2 details the Memory-T1 framework, and Section 3.3 describes the reward design used to train the agent.

## 3.1 PROBLEM FORMULATION

Temporal reasoning in multi-session scenarios is formulated as a QA task (Figure 1): given a user query $q$, the goal is to produce an answer $a$ grounded in the dialogue history. The dialogue history is represented as a memory bank $\mathcal{M} \equiv [(\tau_1, S_1), (\tau_2, S_2), \ldots, (\tau_N, S_N)]$, where each session $S_i$ is associated with a timestamp $\tau_i$ and consists of a sequence of utterances paired with referenced events:

$$S_i = \{(u_{i1}, \mathcal{E}_{i1}), (u_{i2}, \mathcal{E}_{i2}), \ldots, (u_{iL_i}, \mathcal{E}_{iL_i})\}, \tag{1}$$

where $u_{ij}$ denotes the $j$-th utterance in session $i$, and $\mathcal{E}_{ij} = \{e_1, e_2, \ldots, e_K\}$ is the set of events mentioned in that utterance. Each event $e_k$ can be optionally annotate with a semantic descriptor $\kappa_k$ and a temporal span $(t_k^{\text{start}}, t_k^{\text{end}})$ (see Figure 1). These annotations are introduced solely for training-time reward computation and are never accessible during inference. Details of the annotation process are provided in the Appendix A.

## 3.2 MEMORY-T1: TEMPORAL-AWARE MEMORY RETRIEVAL

MEMORY-T1 is a temporal-aware memory retrieval framework designed for multi-session dialogue agents. Its architecture follows a coarse-to-fine filtering principle to efficiently identify relevant and temporally consistent memories from a vast and noisy dialogue history. The process is organized into two main phases: Candidate Generation and Fine-grained Selection.

**Phase 1: Candidate Generation**: This initial phase aims to rapidly prune the large-scale memory repository down to a manageable set of high-recall candidates. It consists of two sequential filtering stages:

1. **Temporal Filtering:** Given a user query q, an LLM first predicts its target temporal window $(t_{\text{start}}, t_{\text{end}})$. This predicted scope acts as a hard filter to discard all sessions whose timestamps do not overlap with this range, drastically reducing the search space and getting temporally-filtered sessions set $M_{\text{temp}}$, which is a subset of the given memory bank $M$ $(M_{\text{temp}} \in M)$.

2. **Relevance Filtering:** From the temporally-filtered sessions, we then use retriever to rank the remaining sessions by textual relevance to the query. This step further narrows the pool to a manageable size that fits within the agent's context budget, while preserving sessions that are both temporally and textually pertinent. The top-ranked sessions form the candidate pool C, formally defined as:

$$\mathcal{C} = \underset{(\tau_i, S_i) \text{ s.t. } t_{\text{start}} \leq \tau_i \leq t_{\text{end}}}{\arg \text{top-}k} \left( \text{Retriever}(q, S_i) \right) \tag{2}$$

**Phase 2: Fine-grained Selection via Reinforcement Learning.** While the candidate set is highly relevant, it may still contain temporally imprecise or misleading information. Reinforcement learning enables the agent to refine its evidence selection policy under reward signals that directly penalize incorrect or temporally inconsistent citations. In this way, RL encourages the model to disambiguate noisy candidates and learn robust mappings between cited evidence and generated answers. Therefore, after identifying the candidate pool $\mathcal{C}$ in Phase 1, we employ an RL-finetuned model to perform the final evidence selection and answer generation in an end-to-end manner.

The agent policy $\pi_\theta$ takes the query q and candidate pool $\mathcal{C}$ as input and generates a single, composite output string. This output is structured to explicitly cite the session IDs used as evidence, followed by the natural language answer. For example, a valid generation would be: $\{selected\_memory : [session\_3, session\_16]. answer : 19 \, days.\}$ From this generated string, we can parse both the selected evidence subset $S \subseteq \mathcal{C}$ (e.g., $[session\_3, session\_16]$) and the final answer $a$. This integrated action space allows the model to learn the direct link between the evidence it cites and the answer it produces. This agent learns a policy $\pi_\theta(S \mid q, \mathcal{C})$ to select a subset of evidence sessions $S$ from the candidate pool $\mathcal{C}$ given the query $q$.

To train this policy, we employ **Group Relative Policy Optimization (GRPO)** (Zheng et al., 2025), an effective RL algorithm for LLM fine-tuning that mitigates high reward variance by using a batch-average baseline. Our overall objective is to maximize the following function:

$$\max_{\theta} J_{\text{GRPO}}(\theta) = \mathbb{E}_{(q,\mathcal{C})\sim\mathcal{D},\{(\mathcal{S}_j,a_j)\}\sim\pi_{\text{ref}}} \left[ \frac{1}{G} \sum_{j=1}^{G} \min\left( r_j(\theta)\hat{A}_j, \text{clip}(r_j(\theta), 1-\epsilon, 1+\epsilon)\hat{A}_j \right) \right]$$
$$- \beta \mathbb{E}_{(q,\mathcal{C})\sim\mathcal{D}} \left[ D_{\text{KL}}\left( \pi_{\theta}(\cdot|(q,\mathcal{C})) \,\|\, \pi_{\text{ref}}(\cdot|(q,\mathcal{C})) \right) \right]. \tag{3}$$

Following a structure similar to PPO (Schulman et al., 2017), we first define a probability ratio $r_k(\theta) = \frac{\pi_{\theta}((\mathcal{S}_j,a_j))|(q,\mathcal{C}))}{\pi_{\text{ref}}((\mathcal{S}_j,a_j))|(q,\mathcal{C}))}$, where $\mathcal{S}_j$ and $a_j$ represent the evident session id set and answer in $j$-th generated output. Here, $\epsilon$ is a clipping hyperparameter that restricts the size of policy updates. The advantage estimate $\hat{A}_j$ corresponding to a sampled generation that yields the pair $(S_j, a_j)$ is calculated against the batch-average reward:

$$\hat{A}((q,C),(\mathcal{S}_j,a_j)) = R((q,C),(\mathcal{S}_j,a_j)) - \frac{1}{G} \sum_{j=1}^{G} R((q,C),(\mathcal{S}_j,a_j)). \tag{4}$$

The reward $R$ is given by a multi-level function in Section 3.3. The second term in Eq. (3) is a KL divergence penalty regularizing the current policy $\pi_{\theta}$ against a frozen reference $\pi_{\text{ref}}$ to ensure training stability. Algorithmic details are in Appendix B.

## 3.3 REWARD DESIGN

In this section, we describe the design of our verifiable rewards. The core motivation of the multi-level reward is to address the limitation of sparse supervision. As shown in Table 1, models such as MemAgent (Yu et al., 2025), which are trained solely on answer accuracy ($R_a$), fail to develop effective temporal reasoning abilities. Thus, it is necessary to jointly optimize evidence grounding ($R_g$, ensuring the correct sessions are used) and temporal consistency ($R_t$, ensuring temporal alignment with query) to form a dense, structured reward signal. Since all rewards assume that the model output can be successfully parsed into the required format (e.g., $\{selected\_memory : .... \ answer : ....\}$), we assign a fixed penalty of $-0.5$ if parsing fails. The overall reward is defined as:

$$R = \begin{cases} w_a R_a + w_g R_g + w_t R_t, & \text{if parsing succeeds,} \\ -0.5, & \text{otherwise,} \end{cases} \qquad R \in [-1,1]. \tag{5}$$

where $w_a, w_g, w_t$ are tunable weights with $w_a + w_g + w_t = 1$. Exact values are in Appendix C.1, and sensitivity to different settings is analyzed in Appendix C.2.

**Accuracy Reward ($R_a$)** This reward ensures that the final predicted answer is correct, providing the most direct supervision signal for the agent's output quality. As tasks require different answer formats, $R_a$ is a multifaceted metric tailored to four main types, each with a specialized evaluation function. For **Option Answers** (e.g., "A", "A C"), we use a strict Exact Match (EM). For numerical answers involving dates or durations, we employ more flexible metrics: **Timestamp** answers (e.g., "2024-09-01") are assessed with Unit-aware Accuracy, while **Time Interval** answers (e.g., "13 days") use $\epsilon$-Exact Match ($\epsilon$-EM). Finally, for sequential answers like **Event Order**, we use Hamming Accuracy to credit partial correctness. The final reward $R_a$ is normalized to the range $[-1, 1]$, with detailed formulations in Appendix C.3.

**Evidence Grounding Reward ($R_g$)** This reward encourages the model to retrieve and utilize information from the correct dialogue session(s). Specifically, this reward is calculated by comparing the set of session IDs $\mathcal{C}$ cited by the agent against the gold-standard evidence set, $M^*$, provided in the dataset. The degree of match is quantified using the Jaccard Index, which measures similarity by dividing the size of the intersection of the two sets by the size of their union. This score is then scaled to range $[-1, 1]$ where a perfect match (Jaccard Index of 1) corresponds to a reward of +1, and a complete mismatch (Jaccard Index of 0) results in a reward of -1.

**Temporal Consistency Reward** ($R_t$): This reward component enforces a fine-grained temporal alignment between the selected sessions and the query. It is composed of two sub-rewards: chronological proximity ($R_s$) and temporal coverage ($R_f$).

$$R_t = \alpha R_s + \beta R_f, \quad (\alpha + \beta = 1) \tag{6}$$

**1. Chronological Proximity ($R_s$, session-level):** This reward measures the temporal distance between the selected session timestamp $U$ and the gold temporal range $I_Q$ of user query. Recognizing that a hard-cutoff penalty is too rigid for the temporal ambiguities in real-world dialogues (e.g., timezone shifts, extended topics), we employ a logistic function to create a soft, differentiable penalty that better handles this imprecision. The reward is formulated as:

$$R_s = \frac{c}{1 + \exp(x)} - d, \quad R_s \in (-d, \, c - d], \tag{7}$$

where the normalized distance $x$ is defined as:

$$x := \frac{\text{gap}(U, I_Q) - m}{s}. \tag{8}$$

Here, $\text{gap}(U, I_Q)$ is the minimum temporal distance (in days) between spans $U$ and $I_Q$ (zero if they overlap). The hyperparameters offer fine-grained control: the tolerance margin $m$ sets a penalty-free grace period (e.g., 7 days), the scale factor $s$ controls the penalty curve sharpness, and the parameters $c, d$ scale the final reward magnitude (to a range $(-0.5, 1]$). This logistic approach ensures that sessions close to $Q$ are highly rewarded while distant sessions are penalized. For detailed settings, please refer to Appendix C.1.

**2. Chronological Fidelity ($R_f$, utterance-level).** While $R_s$ handles session-level relevance, $R_f$ evaluates the fine-grained quality of *events* within each utterance. It rewards sessions dense with evidence that is temporally aligned with the time range of the query, $I_Q$. First, we assign a discrete score $r_e$ to each event $e$ based on its temporal overlap with $I_Q$:

$$r_e(e, I_Q) = \begin{cases} +1, & \text{if the time range of event } e \text{ is fully within } I_Q, \\ +0.5, & \text{if partially overlaps with } I_Q, \\ -1, & \text{if no overlap with } I_Q. \end{cases} \tag{9}$$

The final fidelity reward $R_f$ is then calculated by first averaging these event scores within each relevant utterance ($u \in U_{\text{rel}}$), and then averaging the resulting utterance scores across the session:

$$R_f(U, I_Q) = \begin{cases} \dfrac{1}{|U_{\text{rel}}|} \sum_{u \in U_{\text{rel}}} \left( \dfrac{1}{|E_u|} \sum_{e \in E_u} r_e(e, I_Q) \right), & \text{if } |U_{\text{rel}}| > 0, \\ 0, & \text{otherwise.} \end{cases} \tag{10}$$

This reward structure effectively penalizes a common failure mode: selecting a session from the correct time period but grounding the answer in a textually similar but temporally incorrect utterance from within it. It incentivizes the agent to select sessions that are not just broadly relevant but also *densely packed with chronologically precise evidence*. By combining these three reward signals ($R_a, R_g, R_t$), our multi-level reward structure guides the agent to develop a robust, generalizable temporal reasoning policy that does not overfit to superficial cues.

## 4 EXPERIMENTS

### 4.1 DATASETS

**Time-Dialog** We use Time-Dialog as the core benchmark, extended from the dialogue portion of the existing Time dataset (Wei et al., 2025), containing 4,716 QA examples corresponding with the multi-session dialogue history as shown in Figure 1.o train a robustly time-aware agent, we augment the dataset with fine-grained annotations for supervision, specifically annotating the target time range for each query, utterance-level events with their time spans, and the ground-truth session

IDs for ideal evidence retrieval. Further details are shown in Appendix A. Crucially, these fine-grained annotations serve exclusively as a ground-truth signal for computing our multi-level rewards during training. To ensure a fair and realistic evaluation, this enriched information is withheld from all models during inference. The final dataset of 4,716 examples is partitioned into training (4,065), validation (451), and held-out test (200) sets.

**LoCoMo**    To assess the out-of-domain (OOD) generalization of our trained policy, we employ the LoCoMo benchmark (Maharana et al., 2024), an established testbed for multi-session conversational memory. LoCoMo is composed of five distinct subtasks, one of which is specifically designed to evaluate temporal reasoning. This makes it an ideal held-out test set to validate whether our model has learned a generalizable temporal reasoning skill, rather than overfitting to the patterns of the Time-Dialog dataset.

## 4.2    EXPERIMENTS SETUP

**Baselines**    Our proposed method, MEMORY-T1, is built upon **Qwen2.5-3B** and **Qwen2.5-7B-Instruct** (Team, 2024). We compare it against a comprehensive suite of baselines, including standard methods like **Full Context**, which is evaluated across a wide range of open-source models (**Qwen2.5**-3B/7B/14B, **Gemma-4B-it** (Team, 2025), **LLaMa-3.1-8B-Instruct** (Dubey et al., 2024)) and the closed-source **GPT-4** (Achiam et al., 2023); standard **Retrieval-Augmented Generation (RAG)** (Lewis et al., 2020); the agentic **ReAct** framework using GPT-4 as its backbone (Yao et al., 2023); and a **Supervised Fine-Tuning (SFT)** model fine-tuned from Qwen2.5-3B (Ouyang et al., 2022). Furthermore, we benchmark against two state-of-the-art specialized agents, **MemAgent** (Yu et al., 2025) and **Time-R1** (Liu et al., 2025), by evaluating their public checkpoints in a zero-shot setting. Finally, to isolate the benefits of our contributions, we include an **RL (Task Reward Only)** ablation baseline, which uses the same architecture as MEMORY-T1 but is trained only with a task accuracy reward ($R_a$), omitting our proposed temporal consistency ($R_t$) and evidence grounding ($R_g$) rewards.

**Implementation**    All our experiments build upon Qwen2.5-3B-Instruct and Qwen2.5-7B-Instruct as the main models. We adopt BM25 as retriever model due to the efficiency. We adopt the GRPO training strategy within the VERL framework (Sheng et al., 2024). We implement our RL training with a batch size of 32, a learning rate of $1 \times 10^{-6}$, K=8 rollout responses per prompt, KL coefficient = 0.1, and a maximum sequence length of 16k tokens.

## 4.3    RESULTS

As shown in Table 1, MEMORY-T1 establishes a new state-of-the-art, with our 3B and 7B models achieving top overall scores of 66.9% and 67.0%. This performance represents a significant leap over a diverse set of baselines. Compared to specialized SOTA models, our trained agent surpasses the zero-shot performance of both the temporal reasoning model Time-R1 (49.4%) and the memory-based framework MemAgent (49.9%) by over 17 absolute points, highlighting the necessity of targeted training for this complex task. Crucially, our approach proves superior to simply increasing model scale. Our 3B model not only consistently outperforms larger models from different families, including Gemma-4B (45.0%), Llama-3-8B (48.4%), and even the much larger Qwen2.5-14B (60.7%), but also performs nearly identically to our 7B variant. This strongly suggests that the performance gains stem primarily from our learned policy rather than the scale of the base model. Notably, MEMORY-T1 also outperforms standard GPT-4 configurations, surpassing both Full Prompt (64.8%) and ReAct (62.8%). While a gap remains to the ideal GPT-4 (Oracle) score of 86.2%, this overall dominance confirms that our learned memory policy is both necessary and effective. This advantage is driven by our model's particularly strong performance on complex reasoning tasks, such as order reasoning (OR) and range reasoning (RR), directly validating the effectiveness of its temporally grounded memory selection policy.

## 4.4    ABLATION STUDY

**Ablation Study on reward components.**    Our multi-component reward function is crucial for robust performance, as shown in Table 2. Training with only task accuracy ($R_a$) leads to a catas-

Table 1: Performance comparison across different models and training strategies on temporal reasoning subtasks. **Category A**'s metrics include Location (Loc.), Duration Comparison (DC.), Comparison (Comp.), Order Comparison (OC.), and Extraction (Ext.). **Category B**'s metrics covers ER.=Event Reasoning, OR.=Order Reasoning, RR.=Range Reasoning. **Category C**'s metrics comprises CTF.=Contextual Temporal Filtering, Co-tmp.=Co-temporality, TL.=Timeline. **Bold** and underline denote column-wise best and second-best *among non-GPT rows*. [†]Oracle setting using gold test evidence.

| Experiments | Category A | | | | | Category B | | | Category C | | | Overall |
|---|---|---|---|---|---|---|---|---|---|---|---|---|
| | Loc. | DC. | Comp. | OC. | Ext. | ER. | OR. | RR. | CTF. | Co-tmp. | TL. | |
| GPT-4 (Oracle Evidence)[†] | 88.9 | 78.3 | 54.9 | 95.0 | 66.7 | 100.0 | 88.9 | 93.8 | 83.3 | 100.0 | 35.4 | 86.2 |
| GPT-4 (Full Prompt) | 61.1 | 87.0 | 46.7 | 85.0 | 22.2 | 64.1 | 55.6 | 81.3 | 50.0 | 77.8 | 27.1 | 64.8 |
| GPT-4 (ReAct) | 66.7 | 43.5 | 39.2 | 75.0 | 32.2 | 67.1 | 72.2 | 70.8 | 68.5 | 84.3 | 29.2 | 62.8 |
| Gemma-4B-it | 5.6 | **60.9** | 12.7 | 55.0 | 33.3 | 62.2 | 38.9 | 50.0 | 44.4 | 61.1 | 15.3 | 45.0 |
| Time-R1 | 11.1 | 47.8 | 13.9 | 65.0 | 55.6 | 76.9 | 27.8 | 44.4 | 55.6 | 66.7 | 25.0 | 49.4 |
| Qwen2.5-3B (Instruct) | 5.6 | 56.5 | 7.2 | 70.0 | 44.4 | 66.7 | 33.3 | 56.3 | 55.6 | 77.8 | 12.5 | 49.4 |
| Qwen2.5-3B+SFT | 22.2 | 56.5 | 7.9 | 65.0 | 44.4 | 66.7 | 33.3 | 56.3 | 55.6 | 77.8 | 16.7 | 50.6 |
| **Memory-T1 (3B)** | 50.0 | 52.2 | 7.1 | **75.0** | 55.6 | 82.1 | 66.7 | **87.5** | **88.9** | **94.4** | 12.4 | 66.9 |
| Llama-3-8B (Instruct) | 22.2 | 43.5 | 5.6 | **75.0** | 33.3 | 79.5 | 27.8 | 56.3 | 72.2 | 27.8 | 14.6 | 48.4 |
| MemAgent-7B | 55.6 | 47.8 | 10.2 | 55.0 | 40.7 | 61.5 | 38.9 | 62.5 | 38.9 | 72.2 | **27.1** | 49.9 |
| Qwen2.5-7B (Instruct) | **61.1** | 52.2 | 0.0 | **75.0** | 55.6 | 63.3 | 38.9 | 54.2 | 50.0 | 72.2 | 16.7 | 53.2 |
| Qwen2.5-14B (Instruct) | 16.7 | 47.8 | 4.4 | 70.0 | 55.6 | **84.6** | 66.7 | 75.0 | 69.7 | **94.4** | 20.8 | 60.7 |
| **MemoryT1 (7B)** | **61.1** | 52.2 | 8.6 | 65.0 | **56.7** | 71.8 | **83.3** | **87.5** | **88.9** | **94.4** | **27.1** | **67.0** |

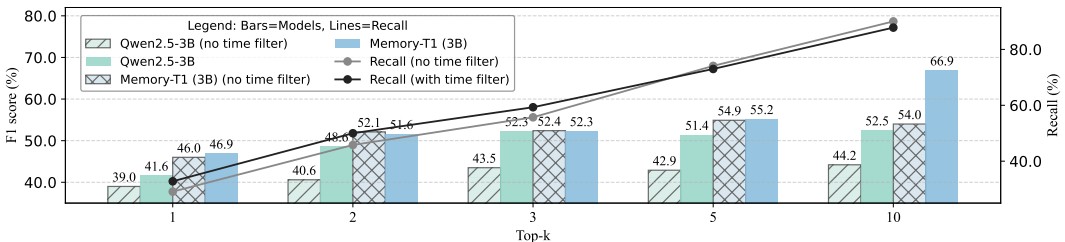

Figure 3: Performance comparison between Memory-T1 (3B) and Qwen2.5-3B (Instruct) under different top-k values (bar charts represent overall F1 scores; line charts represent evidence session recall rate. Comparison conditions: With/without temporal filtering; Top-k refers to the number of sessions retrieved in the candidate generation phase.)

trophic 22.4% drop in the overall score, with performance on complex reasoning (Category B & C) collapsing. Removing the evidence grounding reward ($w/oR_g$) significantly harms localization and extraction-based tasks (Category A, -17.4%), causing a 9.1% overall performance drop and confirming its role in preventing distraction. The temporal consistency reward ($R_t$), composed of sequence ($R_s$) and fine-grained ($R_f$) components, is vital for structured reasoning. Most revealingly, ablating only the sequence component ($-R_s$) creates a sharp trade-off: simpler tasks (Category A) unexpectedly improve by 23.4%, while complex reasoning (Category B) collapses by 56.2%. This highlights a crucial synergy: $R_g$ grounds the model in *what* evidence to use, while $R_t$ teaches it *how* to reason with that evidence temporally. To clarify the non-monotonic effects of $R_t$ components, Category-A duration tasks rely on two complementary mechanisms: global timeline consistency ($R_s$) and content-level temporal relevance ($R_f$). Removing only one leaves the other as a compensatory constraint, improving simpler timestamp- or gap-based reasoning. Full removal of $R_t$ eliminates both regulating factors, preventing correct event selection and temporal alignment, which explains the sharp performance drop.

## 4.5 MODEL ANALYSIS

**Out-of-Domain Generalization.** Our model demonstrates strong out-of-domain (OOD) generalization on the LoCoMo benchmark (Table 3, Table 9 in Appendix C)). MEMORY-T1 achieves a top score of **37.7%**, a significant improvement over the 33.5% from the base Qwen-2.5-3B model. This advantage is particularly consistent in the Non-RAG setting (31.9% → 36.7%), driven by substantial

Table 2: Ablation study on the reward function of Memory-T1 (3B). Relative changes compared to the full model are shown in parentheses.

| Model | Category A | Category B | Category C | Overall |
|---|---|---|---|---|
| **Memory-T1 (3B)** | 49.5 | 79.5 | 80.3 | 66.9 |
| *w/o* $R_t$ | 45.6 (-7.9%) | 75.1 (-5.5%) | 64.3 (-19.9%) | 63.5 (-5.1%) |
| – remove $R_s$ only | 61.1 (+23.4%) | 34.8 (-56.2%) | 66.3 (-17.4%) | 66.3 (-0.9%) |
| – remove $R_f$ only | 50.0 (+1.0%) | 56.5 (-28.9%) | 63.0 (-21.6%) | 64.8 (-3.1%) |
| *w/o* $R_g$ | 40.9 (-17.4%) | 75.3 (-4.2%) | 75.9 (-5.5%) | 60.8 (-9.1%) |
| $R_a$ only | 43.6 (-11.9%) | 57.5 (-27.7%) | 59.0 (-26.6%) | 51.9 (-22.4%) |

**Ablation Study on candidate generation phase.** Figure 3 validates our coarse-to-fine candidate generation strategy. First, increasing retrieval depth top-k to 10 is essential to achieve high evidence recall ( 90%). Our temporal filter proves highly precise, as the overlapping recall lines show it removes distracting context without sacrificing this crucial evidence. Second, even with the same unfiltered context, our RL-tuned MEMORY-T1 agent outperforms the base model (54.0% vs. 44.2%). The synergy of combining broad retrieval for high recall with sharp, evidence-preserving filtering creates an optimal candidate pool that enables the agent to achieve its final 66.9% score.

Table 3: LoCoMo benchmark: Out-of-Domain evaluation of Qwen-2.5-3B-Instruct and Memory-T1 (3B) under RAG and Non-RAG settings. Values are shown as percentages; best results in each column are bolded. ΔOverall shows improvement relative to Qwen-2.5-3B-Instruct (Non-RAG).

| Model Family | Setting | Single-Hop | Multi-Hop | Temporal | Open-Domain | Adversarial | Overall | ΔOverall (%) |
|---|---|---|---|---|---|---|---|---|
| Qwen-2.5-3B (Instruct) | Non-RAG | 49.8 | 28.7 | 24.5 | 13.5 | 16.6 | 33.5 | – |
| | RAG | 46.0 | 22.0 | 27.3 | 11.4 | 19.5 | 31.9 | -1.6% |
| Memory-T1 (3B) | Non-RAG | **51.2** | **30.2** | **31.5** | **15.8** | 26.0 | **37.7** | +4.2% |
| | RAG | 48.9 | 25.8 | 30.7 | 14.6 | **29.8** | 36.7 | +3.2% |

Table 4: Robustness of Memory-T1 under increasing time label noise.

| Noise Level | Category A | | | | | Category B | | | Category C | | | Overall |
|---|---|---|---|---|---|---|---|---|---|---|---|---|
| | Loc. | DC. | Comp. | OC. | Ext. | ER. | OR. | RR. | CTF. | Co-tmp. | TL. | |
| 20% | 27.8 | 43.5 | 5.0 | 55.0 | 25.9 | 76.9 | 72.2 | 81.2 | 94.4 | 94.4 | 16.7 | 60.0 |
| 10% | 50.0 | 43.5 | 10.6 | 65.0 | 55.6 | 74.4 | 67.4 | 81.2 | 88.9 | 94.4 | 18.8 | 63.4 |
| 5% | 50.0 | 60.9 | 5.0 | 60.0 | 55.6 | 82.0 | 77.8 | 87.5 | 94.4 | 88.9 | 16.7 | 67.0 |

gains in the **Temporal** and **Adversarial** subtasks. Intriguingly, MEMORY-T1 yields better performance in the Non-RAG setting compared to the RAG setting, suggesting it has learned a superior internal memory management skill. The **Adversarial** subset is a notable exception, which focuses on answerability detection (saying "I don't know" when the information is missing). Without the RAG setting, the Post-filter candidate pool remains lengthy and is prone to "lost in the middle" effects and spurious snippets that encourage hallucination. With RAG, the condensed candidate pool prunes spurious in-dialog segments, preserving a compact set lacking supporting evidence. This makes it easier for the RL policy to learn "unanswerable" behavior more effectively ($26.0 \rightarrow 29.8$). It introduces a mild distribution shift and loss of temporally key candidates on standard tasks but benefits adversarial detection by simplifying evidence incompleteness detection.

**Robustness in Long-Context Scenarios.** To assess how models handle increasingly complex dialogues, we partition the test set by context length and evaluate performance on each bracket (Figure 4). As context length increases, the performance of baseline models collapses due to attentional dilution; the Qwen2.5-7B baseline, for instance, drops by over 30 F1 points. In contrast, MEMORY-T1 maintains a high and stable F1 score across all lengths. This creates a performance gap that widens dramatically with context, growing from a +9.8 point advantage to a massive **+25.0 point** lead for MEMORY-T1 (7B) in the 64k-128k bracket. This resilience stems directly from our learned policy, which effectively filters context and shields the model from distraction, confirming its superiority for long-range reasoning. Further controlled experiments on lost-in-the-middle effects are provided in Appendix C.5.

**Robustness under increasing time label noise.** As shown in Table 4, with 5% noise (realistic error rate), overall F1 remains 67.0, and key temporal reasoning tasks such as Counterfactual (CTF.),

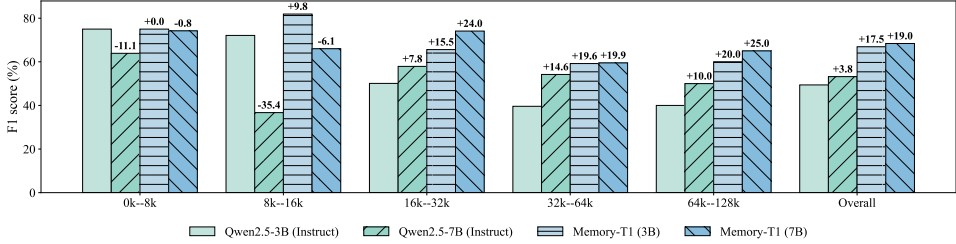

Figure 4: Comparison of Qwen2.5 and Memory-T1 models on the test set, where examples are grouped by the length of each test example (tokens) (0k–8k, 8k–16k, 16k–32k, 32k–64k, 64k–128k) to assess performance variation across lengths, along with overall evaluation.

Table 5: Analysis of model performance and Retrieval-Augmented Generation (RAG) Latency (time in seconds)

| Model | Num of Q | Total Inf. Time | Avg Latency | Total Inf. w R | Retrieval Time |
|---|---|---|---|---|---|
| Time-R1 | 200 | 248.62 | 1.24 | 256.35 | 0.01 |
| MemAgent | 200 | 312.72 | 1.56 | 320.47 | 0.01 |
| Qwen2.5-3B (Instruct) | 200 | 271.83 | 1.36 | 279.74 | 0.01 |
| Memory-T1 | 200 | 252.08 | 1.26 | 259.81 | 0.01 |

Co-temporality (Co-tmp.), and Relative Reasoning (RR.) stay high at 94.4, 88.9, and 87.5, respectively. Increasing the noise to 10% and 20% leads to a gradual but moderate degradation of the overall score to 63.4 and 60.0. Notably, the most temporally demanding tasks remain robust: CTF. and Co-tmp. stay above 88.9 F1 even at 20% noise. The main decline is concentrated in time-span–related subtasks (such as Localization and Extract). This confirms the Memory-T1 is resilient to realistic label noise, supporting its practical applicability in real-world settings where time labels are imperfect.

**Efficiency Analysis** . Memory-T1 incurs negligible additional inference latency (Table 5). The average latency (1.26 seconds per query) is highly comparable to baselines such as Time-R1 (1.24 seconds) and Qwen2.5-3B (1.36 seconds). Crucially, the retrieval overhead (0.01 seconds) is insignificant relative to the total LLM generation latency, confirming that the framework achieves its improved performance with minimal computational cost.

**Qualitative Analysis.** We focused our qualitative analysis on the six subtasks (ER., OR., RR., CTF., Co-tmp., and Loc.) where Memory-T1 exhibits the largest performance gains (Table 10 in Appendix). A consistent pattern emerges across these subtasks: the base model often relies on semantic similarity rather than temporal correctness, which leads to systematic errors such as neglecting time constraints, confusing event order, overlooking co-temporal relations, and failing to incorporate counterfactual adjustments. Memory-T1 mitigates these issues through explicit time-range filtering and RL-based selection that enforces temporal consistency, yielding more accurate localization, ordering, and co-temporality. These qualitative observations align with and explain the performance improvements observed on the six subtasks.

## 5 CONCLUSION

In this work, we introduce **MEMORY-T1**, a novel reinforcement learning framework addressing the critical challenge of temporal reasoning over long, multi-session dialogues. The framework employs a coarse-to-fine strategy, guided by a multi-level reward function that incorporates answer accuracy, evidence grounding, and a temporal consistency signal. This design provides the agent with dense supervision to effectively handle temporal ambiguities and noise. Experiments show that MEMORY-T1 achieves state-of-the-art performance on the Time-Dialog benchmark, enabling a 3B model to outperform a 14B baseline and maintaining strong robustness in dialogue histories up to 128k tokens. This work demonstrates that selecting temporally consistent memory evidence is a critical step toward building more reliable and factually consistent long-term conversational agents.

REPRODUCIBILITY STATEMENT

We are committed to ensuring the transparency and reproducibility of our research. To support this commitment, we will publicly release our annotated dataset and all source code, facilitating future extensions and community research. Comprehensive details of our methodology are provided throughout this paper: the annotation process and prompts are illustrated in Appendix A, Figures 21, 20, and 22; training and evaluation prompts are shown in Figure 23 and Figure 24, respectively. Furthermore, detailed algorithmic procedures can be found in Appendix B. We believe that releasing these assets will lower the barrier for replication, enable fair comparisons, and foster further exploration in this line of research.

ETHICS STATEMENT

The main artifact of this work is the annotated Time-Dialog dataset. To facilitate the process, we develop a dedicated evidence-annotation website (Figure 5) and engage three experienced NLP researchers as annotators. Approximately 200 human hours are devoted to verifying GPT-4–assisted annotations, categorizing error types, and refining the protocol through several iterations. All annotators are properly briefed and held regular discussions to resolve ambiguous cases. Model evaluations are conducted by three trained research assistants, each compensated at $20/hour, which is above the local average. Prior to release, all data underwent rigorous screening to ensure the exclusion of personally identifiable information and offensive content. Both the dataset and code will be publicly released under an MIT license to encourage transparency and community use.

ACKNOWLEDGEMENTS

This work is partially supported by Hong Kong RGC GRF No. 14206324, the National Natural Science Foundation of China 62576120, the Major Key Project of PCL2025A09 and Key Laboratory of Computing Power Network and Information Security, Ministry of Education under Grant No.2024ZD020.

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

# A  DATASET AND ANNOTATIONS

Our experiments are conducted on the **Time** dataset, a comprehensive benchmark for temporal reasoning over long-form dialogues. The dataset features complex dialogue histories and is structured into 3 levels of reasoning difficulty and 11 distinct QA subtasks. The distribution of these subtasks in the dataset is detailed in Table 6.

Table 6: Distribution and characteristics of QA subtasks in the Time dataset, grouped by reasoning level.

| QA Subtask | Format | category | # Samples |
|---|---|---|---|
| Localization | Time Span | A | 381 |
| Duration_Compare | Single Choice | A | 385 |
| Computation | Time Span | A | 390 |
| Order_Compare | Single/Multi Choices | A | 380 |
| Extract | Single/Multi Choices | A | 197 |
| Explicit_Reasoning | Single Choice | B | 363 |
| Order_Reasoning | Single Choice | B | 381 |
| Relative_Reasoning | Single Choice | B | 393 |
| Counterfactual | Single/Multi Choices | C | 398 |
| Co_temporality | Single/Multi Choices | C | 397 |
| Timeline | Event Order | C | 390 |

While the Time dataset provides a strong foundation, it lacks the fine-grained annotations necessary for our reward mechanisms and detailed analysis. To address this, we augmented the dataset with three additional layers of annotations. Our annotation process employed an iterative framework where GPT-4 performed an initial annotation pass, followed by human verification to identify systematic error patterns. These insights were then used to refine the prompts for a final, improved annotation pass, achieving an overall accuracy of over 95%.

**1. Question Temporal Range** ($I_Q$).   First, for each question, we annotate its **target temporal range** ($I_Q$). Many questions implicitly focus on a specific period within the long dialogue history. We prompted GPT-4 to infer and extract this time range. For questions with no discernible temporal focus, we assigned a default range starting from "unknown" to our annotation timestamp (e.g., "2025-07-17T11:46:32"). As this timestamp is later than any event in the dataset, this default range effectively covers the entire dialogue history. The prompt can be found in Figure 21

**2. Evidence Grounding** ($\mathcal{M}^*$).   Second, we annotate the **ground-truth evidence sessions** ($\mathcal{M}^*$) **and utterances** required to answer each question. The original dataset's fact bank could not be reliably mapped to the dialogue text. We therefore used our iterative GPT-4 (Figure 22) and human-in-the-loop process (Figure 5) to perform this grounding. This resulted in a session-level annotation accuracy of over 95% and an utterance-level accuracy of over 85%. To avoid introducing potential noise from less accurate annotations into our reinforcement learning process, we use the more reliable **session-level annotations** for calculating the Evidence Grounding Reward ($R_g$).

**3. Utterance-level Event Times.**   Finally, to enable a deeper temporal analysis, we performed **utterance-level event extraction and temporal grounding** for the entire dialogue history (Figure 20). This annotation is crucial because the timestamp of a dialogue turn (when something was said) often differs from the timestamp of the event being discussed (when something happened). This distinction is the primary motivation for our chronological proximity ($R_f$) reward. For each utterance, we prompted GPT-4 to extract key events and resolve their temporal scope based on the dialogue context. For instance, given a dialogue turn on '2025-06-20', an utterance mentioning "the meeting last week" would be grounded to a specific range like '[2025-06-09, 2025-06-13]'. For utterances without explicit temporal markers, we used grammatical tense to infer a broad range (e.g., past tense implies a range from the distant past up to the dialogue time, while future tense implies a range from the dialogue time to the distant future).

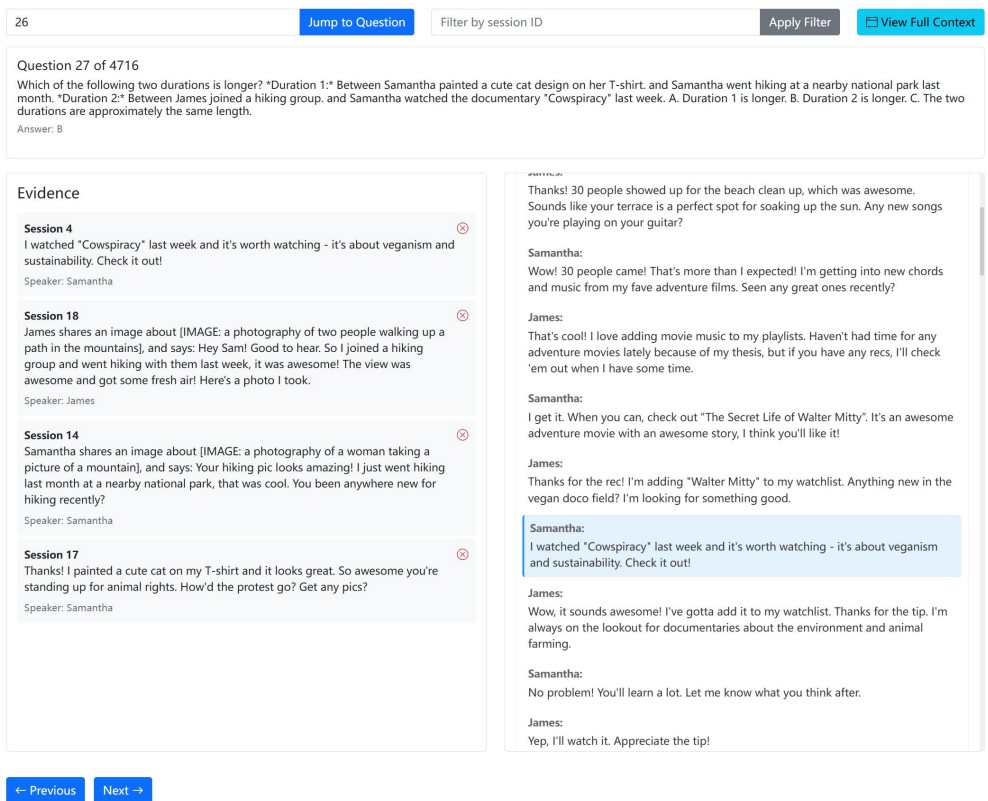

Figure 5: An overview of website for human annotation.

# B ALGORITHM

The core of the framework is a reinforcement learning agent trained with Group Relative Policy Optimization (GRPO) as shown in Algorithm 1. Our overall approach involves a two-phase process: first, an efficient candidate generation stage to prune the search space, followed by a reinforcement learning (RL) fine-tuning stage to train the policy model.

**Phase 1: Candidate Generation.** Given a query $q$ and the full dialogue memory $\mathcal{M}$, we first generate a small, highly relevant pool of candidate sessions $\mathcal{C}$. This step, detailed in Algorithm 2, is crucial for making the subsequent selection process tractable and efficient.

**Phase 2: RL Fine-tuning.** With the candidate set $\mathcal{C}$, we perform an RL update. For each instance in the batch, we sample $G$ distinct outputs from the current policy $\pi_\theta$. Each output contains a selected evidence set $\mathcal{S}_j$ and a generated answer $a_j$. A multi-level reward $R_j$ is then calculated for each of the $G$ samples by comparing it against the ground-truth labels $(\mathcal{M}^*, a^*, I_Q)$. This reward, detailed in Algorithm 3, provides a comprehensive signal reflecting accuracy, evidence grounding, and temporal consistency.

To reduce the variance of the policy gradient estimate, we compute an advantage $\hat{A}_j$ for each sample. Following GRPO, we use a simple yet effective batch-average baseline, where the advantage is the sample's reward minus the average reward across all $G$ samples in the batch ($\hat{A}_j = R_j - \bar{R}$).

Finally, the policy model's parameters $\theta$ are updated using the GRPO objective function. This objective maximizes the advantage-weighted log-probability of the sampled outputs. Crucially, it also includes a Kullback-Leibler (KL) divergence term, $D_{\mathrm{KL}}(\pi_\theta \| \pi_{\mathrm{ref}})$, weighted by $\lambda$. This term regularizes the policy update, preventing the trained policy $\pi_\theta$ from deviating too drastically from a frozen reference policy $\pi_{\mathrm{ref}}$, which is essential for maintaining training stability.

### B.1 CANDIDATE GENERATION (ALGORITHM 2)

The candidate generation process is a critical filtering cascade designed to efficiently narrow down the vast memory repository $\mathcal{M}$ to a small set of promising candidates $\mathcal{C}$. This is achieved through a two-stage process:

**1. Temporal Filtering.** First, we leverage a powerful LLM to perform a zero-shot prediction of the likely temporal window $(t_{\text{start}}, t_{\text{end}})$ relevant to the user query $q$. We then perform an initial broad-phase filtering by retaining only those sessions $(\tau_i, S_i)$ from $\mathcal{M}$ whose timestamps $\tau_i$ overlap with this predicted window. This step effectively prunes the majority of irrelevant sessions based on a strong temporal heuristic.

**2. Relevance Filtering.** The temporally-filtered subset $\mathcal{M}_{\text{temp}}$ is then passed to a second filtering stage. Here, we use a fast and effective lexical retrieval method, BM25, to rank all sessions in $\mathcal{M}_{\text{temp}}$ based on their textual relevance to the query $q$. The final candidate pool $\mathcal{C}$ is formed by selecting the top-ranked sessions from this list. This cascade approach—using a temporal heuristic followed by lexical matching—allows for an efficient and effective reduction of the search space without relying on expensive semantic models at a large scale.

### B.2 MULTI-LEVEL REWARD CALCULATION (ALGORITHM 3)

To provide a rich and informative learning signal for our policy, we designed a multi-level reward function that captures three critical aspects of the task. The final reward $R$ is a weighted sum of these components.

**1. Task-level Accuracy Reward ($R_a$).** This is a sparse, binary reward that directly measures task success. It yields a reward of 1 if the generated answer $a$ is correct with respect to the ground-truth answer $a^*$, and 0 otherwise. This component ensures the model is strongly incentivized to produce factually correct final answers.

**2. Evidence Grounding Reward ($R_g$).** This component evaluates the quality of the retrieved evidence. We calculate the F1-score between the set of session IDs in the predicted evidence set $\mathcal{S}$ and the ground-truth evidence set $\mathcal{M}^*$. This dense reward encourages the model to select the precise set of sessions required to formulate the answer, promoting better interpretability and faithfulness.

**3. Temporal Consistency Reward ($R_t$).** This novel reward component assesses the temporal quality of the selected evidence $\mathcal{S}$ with respect to the ground-truth temporal range $I_Q$. It is computed as the average of individual rewards over all selected sessions. For each session $U \in \mathcal{S}$, the reward is a weighted sum of two sub-components:

- **Chronological Proximity ($R_s$):** This measures the temporal distance between the session's timestamp $U$ and the gold range $I_Q$. It uses a logistic function to provide a soft, differentiable penalty, rewarding close proximity and penalizing distant sessions.
- **Chronological Fidelity ($R_f$):** This provides a more fine-grained signal. Within a given session $U$, it assesses whether the events in the utterances are relevant to the query are themselves temporally aligned with the gold range $I_Q$. It returns a positive reward for relevant utterances inside $I_Q$, a smaller positive reward for those on the boundary, and a negative penalty for those outside.

The final reward $R$ is the weighted sum $w_a R_a + w_g R_g + w_t R_t$, where the weights allow us to balance the relative importance of task accuracy, evidence quality, and temporal alignment.

---

**Algorithm 1** MEMORY-T1 Training Procedure

---

**Require:**
    Full dialogue memory repository $\mathcal{M}$
    Training dataset $\mathcal{D} = \{(q_i, \mathcal{M}_i^*, a_i^*, I_{Qi})\}_{i=1}^N$, where $\mathcal{M}^*$ is the ground-truth evidence set, $a^*$ is the ground-truth answer, and $I_Q$ is the ground-truth temporal range
    Policy model to be trained $\pi_\theta$ and a frozen reference policy $\pi_{\text{ref}}$
    Hyperparameters: KL divergence weight $\lambda$, reward weights $w_a, w_g, w_t$, group size $G$
**Ensure:**
    Optimized policy model $\pi_{\theta'}$
1: **function** TRAINMEMORY-T1($\mathcal{M}, \mathcal{D}, \pi_\theta, \pi_{\text{ref}}$)
2:     Initialize policy parameters $\theta$
3:    **for** each training iteration **do**
4:       Sample a batch $(q, \mathcal{M}^*, a^*, I_Q)$ from $\mathcal{D}$
    ▷ *Phase 1: Candidate Generation*
5:       $\mathcal{C} \leftarrow$ GenerateCandidates($q, \mathcal{M}$)                       ▷ Call Algorithm 2
    ▷ *Phase 2: RL Fine-tuning*
6:       SampledOutputs $\leftarrow$ []
7:       **for** $j = 1$ **to** $G$ **do**                     ▷ Sample K outputs from the policy
8:          Generate an output string from $\pi_\theta$ conditioned on $(q, \mathcal{C})$
9:          Parse the selected evidence set $\mathcal{S}_j$ and answer $a_j$ from the string
10:         Add $(\mathcal{S}_j, a_j)$ to SampledOutputs
11:       **end for**
12:       Rewards $\leftarrow$ []
13:       **for** $j = 1$ **to** $G$ **do**               ▷ Calculate reward for each sample
14:          $R_k \leftarrow$ CalculateReward($(\mathcal{S}_j, a_j), (\mathcal{M}^*, a^*, I_Q)$)     ▷ Call Algorithm 3
15:          Add $R_j$ to Rewards
16:       **end for**
17:       **for** $j = 1$ **to** $G$ **do**                    ▷ Calculate Advantage
18:          $\hat{A}_j \leftarrow R_j - \frac{1}{G}\sum_{j=1}^g R_j$         ▷ GRPO's batch-average baseline
19:       **end for**
    ▷ *Policy Update*
20:       Update model parameters $\theta$ using the GRPO objective:
21:       $\nabla_\theta J(\theta) \approx \sum_{j=1}^G \nabla_\theta \log \pi_\theta((\mathcal{S}_j, a_j) \mid (q, \mathcal{C}) \hat{A}_j - \lambda \nabla_\theta D_{\text{KL}}(\pi_\theta \,\|\, \pi_{\text{ref}})$
22:    **end for**
23:    **return** the trained policy model $\pi_{\theta'}$
24: **end function**

---

## C   REWARD SUPPLEMENTARY

### C.1   HYBERPARAMETER DESIGN

The Temporal Consistency Reward $R_t = \alpha R_s + \beta R_f$ and its component $R_s$ are governed by a set of hyperparameters $(c, d, m, s, \alpha, \beta)$. The function of each parameter group remains the same: **Tolerance and Leniency** $(m, s)$ defines the "softness" of the temporal alignment. The tolerance margin $(m)$ sets a grace period, while the scale factor $(s)$ control the sharpness of the penalty curve outside this margin. **Incentive Scaling** $(c, d)$**:** controls the magnitude of the reward and penalty, allowing us to calibrate the strength of the positive and negative incentives. **Component Weighting** $(\alpha, \beta)$**:** balance the importance of chronological proximity $(R_s)$ versus chronological fidelity $(R_f)$.

### C.2   SENSITIVE ANALYSIS

To assess the sensitivity of the model to the composition of the reward function, we evaluate its performance under four different weight configurations for accuracy $(w_a)$, evidence grounding $(w_g)$, and temporal consistency $(w_t)$, with results in Figure 6. We find that optimal performance (67.0%) is achieved with the configuration $(0.6, 0.2, 0.2)$. This result suggests that while task accuracy is the primary objective, substantial weights for both evidence grounding and temporal consistency are essential to guide the reasoning process of the agent effectively. Deviating from this balance leads

---

**Algorithm 2** Candidate Generation

---

**Require:**
    User query $q$
    Full dialogue memory repository $\mathcal{M} = [(\tau_1, S_1), \ldots, (\tau_N, S_N)]$
**Ensure:**
    Candidate session pool $\mathcal{C}$
  1: **function** GENERATECANDIDATES($q, \mathcal{M}$)
      $\triangleright$ *1. Temporal Filtering*
  2:    Predict target temporal window $(t_{\text{start}}, t_{\text{end}})$ for query $q$ using an LLM
  3:    $\mathcal{M}_{\text{temp}} \leftarrow \emptyset$
  4:    **for** each session $(\tau_i, S_i)$ in $\mathcal{M}$ **do**
  5:        **if** timestamp $\tau_i$ overlaps with $(t_{\text{start}}, t_{\text{end}})$ **then**
  6:            $\mathcal{M}_{\text{temp}} \leftarrow \mathcal{M}_{\text{temp}} \cup \{(\tau_i, S_i)\}$
  7:        **end if**
  8:    **end for**
      $\triangleright$ *2. Relevance Filtering*
  9:    Rank all sessions in $\mathcal{M}_{\text{temp}}$ by textual relevance to query $q$ using BM25
10:    $\mathcal{C} \leftarrow$ Select top-ranked sessions from the sorted list
11:    **return** $\mathcal{C}$
12: **end function**

---

Table 7: Heuristic configuration of hyperparameters for the temporal reward function.

| Parameter(s) | Value | Rationale |
|---|---|---|
| $c, d$ | 1.5, 0.5 | Normalizes the maximum reward $(c - d)$ to 1, bounding the reward $R_s$ to the range $(-0.5, 1]$. This provides a strong positive signal for a perfect match and a moderate penalty for distant ones. |
| $m$ | 7 (days) | Based on the domain knowledge that a one-week window is a reasonable span for contextual relevance in conversational data. |
| $s$ | 1 | Set to a default value to create a standard and predictable logistic decay curve without excessive sharpness or leniency. |
| $\alpha, \beta$ | 0.5, 0.5 | Establishes a robust baseline by giving equal importance to the two sub-rewards: chronological proximity $(R_s)$ and chronological fidelity $(R_f)$. |
| $w_a, w_g, w_t$ | 0.6, 0.2, 0.2 | Selected based on extensive experiments to balance accuracy, evidence grounding, and temporal consistency. |

to a clear degradation in performance. An accuracy-skewed weighting of $(0.8, 0.1, 0.1)$ diminishes the influence of our guiding rewards, causing the score to drop to 64.0%. Similarly, a uniform distribution $(\frac{1}{3}, \frac{1}{3}, \frac{1}{3})$ proves suboptimal (62.2%), likely because it fails to sufficiently prioritize the main task goal. These findings underscore that the reward components are synergistic; peak performance hinges on a careful balance rather than maximizing any single objective in isolation.

## C.3 ACCURACY REWARD ($R_a$) METRICS

The Accuracy Reward ($R_a$) evaluates the correctness of the final answer, tailored for four main types. Each metric yields a $Score$ in the range $[0, 1]$, which is then normalized to the final reward $R_a \in [-1, 1]$.

**Option Answers (Exact Match, EM)** For categorical answers (e.g., "A", "A C"), we use a strict Exact Match. The score is defined as $\text{Score}_{\text{EM}} = \mathbb{I}(A_{\text{pred}} = A_{\text{gold}})$, where $\mathbb{I}(\cdot)$ is the indicator

---

**Algorithm 3** Multi-Level Reward Calculation

---

**Require:**
    Generated evidence and answer $(\mathcal{S}, a)$
    Ground-truth evidence, answer, and temporal range $(\mathcal{M}^*, a^*, I_Q)$
    Reward weights $w_a, w_g, w_t, \alpha, \beta$
**Ensure:**
    Total scalar reward $R$
 1: **function** CALCULATEREWARD$((\mathcal{S}, a), (\mathcal{M}^*, a^*, I_Q))$
       ▷ *1. Task-level Accuracy Reward ($R_a$)*
 2:    $R_a \leftarrow (1$ if $a$ is correct w.r.t. $a^*$ else $0)$
       ▷ *2. Evidence Grounding Reward ($R_g$)*
 3:    $R_g \leftarrow$ F1_score(session_ids$(\mathcal{S})$, session_ids$(\mathcal{M}^*))$
       ▷ *3. Temporal Consistency Reward ($R_t$)*
 4:    $R_{t,\text{total}} \leftarrow 0$
 5:    **if** $\mathcal{S}$ is not empty **then**
 6:        **for** each selected session $U \in \mathcal{S}$ **do**
          ▷ *a. Chronological Proximity ($R_s$)*
 7:        $x \leftarrow (\text{gap}(U, I_Q) - m)/s$
 8:        $R_s \leftarrow c/(1 + \exp(\kappa x)) - d$
          ▷ *b. Chronological Fidelity ($R_f$)*
 9:        $U_{\text{rel}} \leftarrow \{u \in U \mid \text{text\_similarity}(u, q) > \text{threshold}\}$
10:        **if** $|U_{\text{rel}}| > 0$ **then**
11:          $R_f \leftarrow \frac{1}{|U_{\text{rel}}|} \sum_{u \in U_{\text{rel}}} \left( \frac{1}{|E_u|} \sum_{e \in E_u} r_e(e, I_Q) \right)$   ▷ $r_e$ scores events (+1, +0.5, -1)
    based on overlap
12:        **else**
13:          $R_f \leftarrow 0$
14:        **end if**
15:        $R_t(U, I_Q) \leftarrow \alpha R_s + \beta R_f$
16:        $R_{t,\text{total}} \leftarrow R_{t,\text{total}} + R_t(U, I_Q)$
17:        **end for**
18:        $R_t \leftarrow R_{t,\text{total}}/|\mathcal{S}|$                                ▷ Average over all selected sessions
19:    **else**
20:        $R_t \leftarrow 0$
21:    **end if**
       ▷ *Combine for final reward*
22:    $R \leftarrow w_a R_a + w_g R_g + w_t R_t$
23:    **return** $R$
24: **end function**

---

function. For instance, if the gold answer $A_{\text{gold}}$ is "B", a prediction of "B" scores 1, while "C" scores 0.

**Timestamp Answers (Unit-aware Accuracy)** To handle various date/time formats, this metric compares canonical representations. Both prediction and ground truth are normalized via a function $N(\cdot)$ before comparison, making the score robust to format differences:

$$\text{Score}_{\text{UnitAware}}(A_{\text{pred}}, A_{\text{gold}}) = \mathbb{I}(N(A_{\text{pred}}) = N(A_{\text{gold}}))$$

For example, a prediction of "2025-09-24" correctly matches the gold answer "September 24, 2025," as both normalize to the same value, yielding a score of 1.

**Time Interval Answers ($\epsilon$-Exact Match, $\epsilon$-EM)** For numerical durations, this metric allows a tolerance $\epsilon$ for minor calculation differences. The score is 1 if the absolute difference between the predicted value $V(A_{\text{pred}})$ and the gold value $V(A_{\text{gold}})$ is within this tolerance:

$$\text{Score}_{\epsilon\text{-EM}}(A_{\text{pred}}, A_{\text{gold}}) = \mathbb{I}(|V(A_{\text{pred}}) - V(A_{\text{gold}})| \leq \epsilon)$$

For a gold answer of "13 days" and with $\epsilon = 1$, predictions from "12" to "14 days" are considered correct.



Figure 6: Sensitive analysis. Heatmap of level-wise performance. $W_1$, $W_2$, $W_3$, $W_4$ correspond to reward weights combination $(w_a, w_g, w_t) = (0.6, 0.2, 0.2)$, $(0.5, 0.25, 0.25), (0.8, 0.1, 0.1), (\frac{1}{3}, \frac{1}{3}, \frac{1}{3})$.

Table 8: PPO vs. GRPO: F1 performance on Memory-T1 models of different sizes.

| Model | Category A | Category B | Category C | Overall |
|---|---|---|---|---|
| **3B, GRPO** | 49.5 | 79.5 | 80.3 | 66.9 |
| **3B, PPO** | 41.2 (-16.8%) | 61.7 (-22.4%) | 68.7 (-14.4%) | 54.5 (-18.5%) |
| **3B, GRPO** | 49.9 | 78.1 | 82.4 | 67.0 |
| **3B, PPO** | 54.5 (+9.2%) | 55.9 (-28.4%) | 57.2 (-30.6%) | 55.6 (-17.0%) |

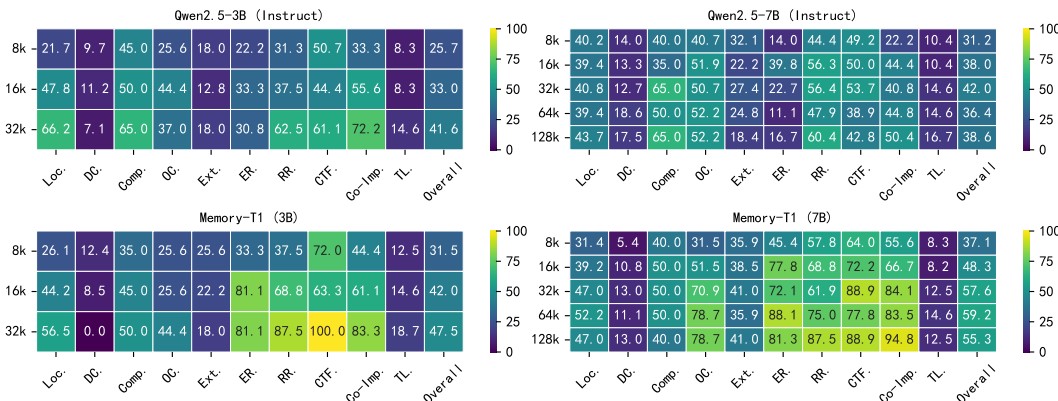

Figure 7: The impact of memory context length on temporal reasoning: F1 performance comparison of Qwen2.5 models and Memory-T1 across context windows of 8k, 16k, 32k, 64k and 128k tokens (retaining the nearest context to the query).

**Sequential Answers (Hamming Accuracy)** To award partial credit for ordered lists, we use Hamming Accuracy, which is the fraction of correctly positioned items. For a prediction $A_{\text{pred}} = (p_1, \ldots, p_L)$ and gold sequence $A_{\text{gold}} = (g_1, \ldots, g_L)$ of length $L$, the score is:

$$\text{Score}_{\text{Hamming}}(A_{\text{pred}}, A_{\text{gold}}) = \frac{1}{L} \sum_{i=1}^{L} \mathbb{I}(p_i = g_i)$$

For example, if $A_{\text{gold}}$ is "(1), (3), (2), (4)" and $A_{\text{pred}}$ is "(2), (3), (1), (4)", the score is $\frac{1}{2}$, as only the second item and fourth item are correct.

**Final Reward Normalization** The final reward $R_a$ is designed to strongly penalize completely incorrect answers while directly rewarding any degree of correctness. If an answer is entirely wrong (Score = 0), it receives a reward of -1. For any partially or fully correct answer (Score > 0), the reward is equal to the score itself. This is formulated as:

$$R_a = \begin{cases} -1 & \text{if Score} = 0 \\ \text{Score} & \text{if Score} > 0 \end{cases}$$

## C.4 COMPARATIVE PERFORMANCE OF GRPO AND PPO

Compared with GRPO, PPO generally underperformed across most categories (Figure 8). For the 3B models, PPO showed substantial declines relative to GRPO, with reductions of -16.8% in Category

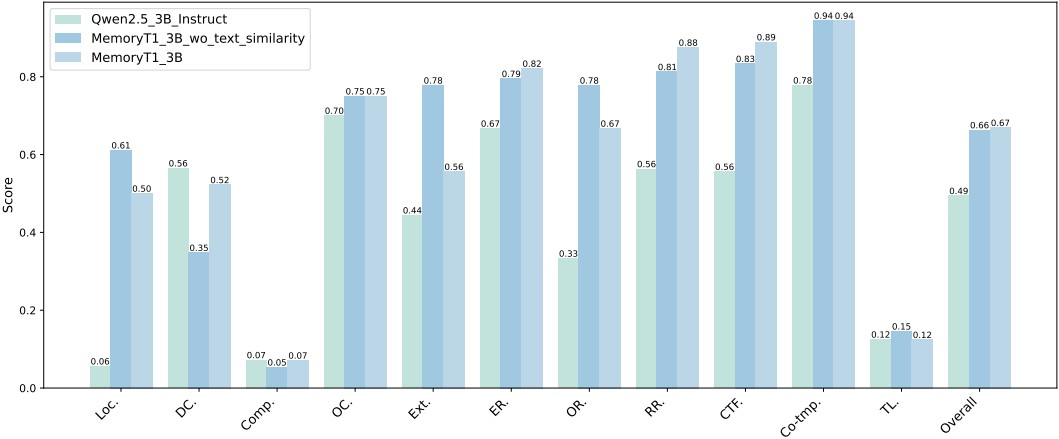

Figure 8: Impact of text similarity filtering component in chronological fidelity reward.

A, -22.4% in Category B, -14.4% in Category C, and -18.5% overall. For the 7B models, PPO achieved a modest improvement in Category A (+9.2%), but suffered marked decreases in Category B (-28.4%) and Category C (-30.6%), leading to a -17.0% drop overall. These results indicate that GRPO provides more stable gains across categories, whereas PPO is less consistent, especially for more challenging tasks (Category B and C).

### C.5 Temporal Reasoning under Controlled Context Windows.

To investigate how the "lost-in-the-middle" problem (Liu et al., 2024; Wang et al., 2025) affects temporal reasoning, we conduct a controlled experiment truncating the input context at various window lengths (Figure 7). The baseline model peaks at a "sweet spot" (e.g., 32k tokens) and then collapses as the context grows longer, a failure caused by attentional dilution. In contrast, our MEMORY-T1 framework is completely resilient to this effect. This resilience stems from our **coarse-to-fine candidate generation**, which filters the noisy history down to a concise and relevant evidence set. By shielding the final agent from irrelevant context, our framework maintains high and stable performance even when the original context exceeds 128k tokens. With this clean, high-quality context, the specific impact of the RL-tuned agent becomes clear. The fine-tuning is highly targeted: it enables the agent to achieve near-perfect "mastery" on specific complex reasoning tasks, with MEMORY-T1(7B) exceeding 0.9 F1 scores on Order Reasoning (OR), Range Reasoning (RR), and Contextual Temporal Filtering (CTF). Conversely, the near-zero scores on Comparison (Comp) and Timeline (TL) highlight the limitations of the current agent paradigm on tasks requiring deeper compositional logic. Finally, we observe a synergy between model scale and fine-tuning, with the RL policy acting as a more powerful performance multiplier on the more capable 7B base model.

### C.6 Impact on text similarity in reward

Ablation results, as shown in Figure 8 clearly demonstrate the pivotal role of the text similarity reward. When this component is present, the model learns to filter out irrelevant dialogue history, thereby anchoring temporal spans more precisely and improving performance on duration-sensitive subtasks. Once the similarity reward is removed, performance on duration computation (DC) and compositional reasoning (Comp.) drops sharply, indicating that the model struggles to maintain accurate temporal spans without explicit guidance to suppress noise. Although slight gains appear in tasks such as Loc. and Ext., these are outweighed by the decline in precision-dependent metrics. This suggests that text similarity primarily functions as a noise-reduction mechanism, ensuring that the reasoning process remains grounded in relevant context, which is especially critical for complex temporal reasoning tasks.

Table 9: MemAgent, Time-R1 model performance comparison: RAG vs. Non-RAG Settings

| Model Family | Params | Setting | F1 Score | Setting | F1 Score |
|---|---|---|---|---|---|
| Time-R1 | 3B | RAG | 31.4 | Non-RAG | 29.2 |
| MemAgent | 7B | RAG | 37.6 | Non-RAG | 40.2 |
| **Memory-T1** | 3B | RAG | 36.7 | Non-RAG | 37.7 |

## C.7 SUPPLEMENTARY EXPERIMENTS: OUT-OF-DOMAIN GENERALIZATION

Memory-T1 (3B) demonstrates strong OOD generalization (Table 9), achieving 37.7% (Non-RAG), which is a significant improvement over Time-R1 (29.2%) and nearly matches the larger MemAgent (7B) (40.2%). This high performance, particularly in the Non-RAG setting, suggests that Memory-T1's learned policy provides a superior internal memory management and reasoning skill that is highly effective and robustly generalizable across domains, outweighing the benefit of RAG observed in other baselines.

## D LLM USAGE

We utilized large language models to support both manuscript polishing and data annotation. In particular, the GPT-4o API is employed to assist with the annotation of the Time-Dialog dataset. Further details of this process are provided in Appendix A.

---

**Localization**

**Type:** Localization
**Format:** time_span
**Level:** level_1
**Question** When is Debra Ryan working on starting her own business?
**Options:** –
**Answer:** 8:35 pm, February 21, 2020

Figure 9: Localization subtask.

---

**Duration Comparison**

**Type:** Duration_Compare
**Format:** single_choice
**Level:** level_1
**Question** Which of the following two durations is longer?
**Options:** A. Duration 1 is longer.
B. Duration 2 is longer.
C. The two durations are approximately the same length.
**Answer:** A

Figure 10: Duration Comparison subtask.

Computation

**Type:** Computation
**Format:** time_span
**Level:** level_1
**Question** How long was it between Debra Ryan going skydiving and India Brown attending a street art fest in Brazil?
**Options:** –
**Answer:** 19 days

Figure 11: Computation subtask.

Order Comparison

**Type:** Order_Compare
**Format:** single_choice
**Level:** level_1
**Question** For Fact1: India Brown became a Queen fan. and Fact2: India Brown found flowers by a lake in the park., which one happened earlier?
**Options:** A. Fact 1 happened earlier.
B. Fact 2 happened earlier.
C. They happen at almost the same time.
**Answer:** A

Figure 12: Order Comparison subtask.

Extract

**Type:** Extract
**Format:** single_choice
**Level:** level_1
**Question** Which of the following are time expressions mentioned in the context?
**Options:** A. April 17, 2021
B. 2018
C. March 16, 2020
D. March 14, 2019
**Answer:** C

Figure 13: Extract subtask.

Explicit Reasoning

**Type:** Explicit_Reasoning
**Format:** single_choice
**Level:** level_2
**Question** What notable artistic or outdoor activities did India Brown participate in between April 1, 2020, and April 9, 2020?
**Options:** A. India Brown attended a street art fest in Brazil.
B. India Brown took a photo of a feather and shells on a beach.
C. India Brown went hiking and sketching at a nearby national park.
D. India Brown received positive feedback on her artwork.
**Answer:** B

Figure 14: Explicit reasoning subtask.

**Order Reasoning**

**Type:** Order_Reasoning
**Format:** single_choice
**Level:** level_2
**Question** What was India Brown's third teaching engagement in 2020?
**Options:** A. Running a painting workshop for kids.
B. Teaching art at an orphanage in Cambodia.
C. Conducting a live demonstration for her college art club.
D. Instructing a pottery class at a local studio.
**Answer:** A

Figure 15: Order reasoning subtask.

**Relative Reasoning**

**Type:** Relative_Reasoning
**Format:** single_choice
**Level:** level_2
**Question** What was India Brown's most recent job before 12:00 am, March 09, 2020?
**Options:** A. New series of abstract artworks.
B. Travel guide based on her trip experiences.
C. New painting technique from street art festival.
D. Testing watercolors for her new series.
**Answer:** A

Figure 16: Relative reasoning subtask.

**Counterfactual**

**Type:** Counterfactual
**Format:** single_choice
**Level:** level_3
**Question** What notable artistic or outdoor activities did India Brown participate in between April 1, 2020, and April 9, 2020, if she visited the Louvre in Paris in March 2020?
**Options:** A. Mini soap sculpture.
B. Photo of a feather and shells.
C. Photograph in Santorini, Greece.
D. Sketched a waterfall during a hike.
**Answer:** B

Figure 17: Counterfactual reasoning subtask.

**Co-temporality**

**Type:** Co_temporality
**Format:** single_choice
**Level:** level_3
**Question** At the same time as Debra Ryan is learning to play the guitar, what collection does India Brown have?
**Options:** A. Soap sculptures.
B. Watercolor paintings.
C. CDs.
D. Vinyl records.
**Answer:** C

Figure 18: Co-temporality subtask.

Table 10: Qualitative analysis of subtasks showing significant improvement (over 10%) in Memory-T1. (e.g., Qwen2.5-3B Instruct Model: (Loc.): $0.278 \rightarrow 0.500$, (ER.): $0.692 \rightarrow 0.821$, (OR.): $0.333 \rightarrow 0.667$, (RR.): $0.563 \rightarrow 0.875$, (CTF.): $0.556 \rightarrow 0.889$, (Co-tmp.): $0.778 \rightarrow 0.944$)

| Subtask | Question | Options | Answer Qwen2.5-3B (Wrong) | Answer Memory-T1 (Correct) |
|---|---|---|---|---|
| Loc. | When is Debra Ryan starting her own business? | N/A | 9:32 pm, May 20, 2020 | 8:35 pm, February 21, 2020 |
| ER. | What notable artistic or outdoor activities did India Brown participate in between April 1, 2020, and April 9, 2020? | A. India Brown attended a street art fest in Brazil. B. India Brown took a photo of a feather and shells on a beach. C. India Brown went hiking and sketching at a nearby national park. D. India Brown received positive feedback on her artwork. | B | C |
| OR. | What was India Brown's third teaching engagement in 2020? | A. Running a teaching workshop for kids. B. Teaching art at an orphanage in Cambodia. C. Conducting a live demonstration for her college art club. D. Instructing a pottery class at a local studio. | B | A |
| RR. | What was India Brown's most recent job before 12:00 am, March 09, 2020? | A. India Brown is working on a new series of abstract artworks based on her trip. B. India Brown is working as a travel guide based on her trip experiences. C. India Brown is working on a new painting technique learned at a street art festival. D. India Brown is testing watercolors for her new series of abstract artworks. | B | A |
| CTF. | What notable artistic or outdoor activities did India Brown participate in between April 1, 2020, and April 9, 2020, if she visited the Louvre in Paris in March 2020? | A. India Brown carved a mini sculpture from a soap bar. B. India Brown took a photo of a feather and shells on a beach. C. India Brown took a photograph in Santorini, Greece. D. India Brown sketched a waterfall during a hike. | C | B |
| Co-tmp. | At the same time as Debra Ryan is learning to play the guitar, what collection does India Brown have? | A. India Brown has a collection of watercolor paintings. B. India Brown has a collection of watercolor paintings. C. India Brown has a collection of CDs. D. India Brown has a collection of vinyl records. | B | C |

Timeline

**Type:** Timeline
**Format:** event_order
**Level:** level_3
**Question** Below are 8 facts. You need to sort these facts in chronological order.
**Options:** (1) New painting technique. (2) Shared mural image. (3) First art show. (4) Became a Queen fan. (5) Invited to exhibit. (6) Beach photo. (7) Sketched waterfall. (8) Received feedback.
**Answer:** (4)(5)(1)(7)(6)(2)(8)(3)

Figure 19: Timeline subtask.

---

**Prompt for Event Extraction and Time Coverage Annotation**

You are a precise temporal reasoner that analyzes utterances in a multi-turn dialogue. Your goal is to analyze each **individual utterance**, based on its content and prior dialogue history, and extract:

- One or more **events** described in the utterance
- For each event:
    - A short summary of the **event** being described
    - The estimated time range of that event
    - The recurring pattern (if applicable) of that event

**Input:**

- Session start time: {`session_start_time`}
- Dialogue history: {`dialogue_history`}
- Current utterance: {`target_utterance`}
- Current speaker: {`speaker`}

**Reasoning Rules:**

- **Event Time Range Estimation:**
    - Explicit date (e.g., "August 14"): use full-day range → start: 00:00:00, end: 23:59:59
    - "yesterday": the day before the utterance time
    - "last week": 7 days ending 1 day before the utterance time
    - Past tense, no time mentioned: start = "unknown", end = utterance time
    - Future tense: start = utterance time, end = "unknown"
    - Habitual/ongoing action: start = "unknown", end = "unknown", mark recurrence
- **Recurring Field:** choose from `none` (default), `daily`, `weekly`, `monthly`, `yearly`, `habitual`

**Output Format (JSON):**

```
[
  {
    "speaker": "Debra Ryan",
    "utterance": "I took this photo last week.",
    "event_summary": "Debra took a photo",
    "event_time":["2020-02-01T00:00:00","2020-02-07T23:59:59"],
    "recurring": "none"
  },
  {
    "speaker": "Debra Ryan",
    "utterance":"I met a friend who was visiting
    from out of town.",
    "event_summary": "Debra met a visiting friend",
    "event_time": ["2020-02-01T00:00:00",
    "2020-02-07T23:59:59"],
    "recurring": "none"
  }
]
```

Ensure your output is valid JSON. Only output the JSON, no extra text.

Figure 20: Prompt used for event extraction and temporal coverage annotation.

---

**Prompt for Question-based Event Reasoning**

You are a precise temporal reasoner that analyzes user's question. You are **given the user's question**.

**Input:**

- User's question: {user_question}

**Reasoning Rules:**

- Event Time Range Estimation:
    - Explicit date (e.g., "August 14"): use full-day range → start: 00:00:00, end: 23:59:59
    - "yesterday": the day before the utterance time
    - "last week": 7 days ending 1 day before the utterance time
    - Past tense, no time mentioned: start = "unknown", end = {current_time_str}
    - Future tense: start = {current_time_str}, end = "unknown"
    - Habitual/ongoing action: start = "unknown", end = "unknown", mark recurrence
- Recurring Field: choose from none (default), daily, weekly, monthly, yearly, habitual

**Output Format (JSON):**

```
{
  "question": "What creative or social activities did
  India Brown participate in between April 16, 2020,
  at 06:22 and April 19, 2020, at 07:22?",
  "time_range": ["2020-04-16T06:22:00", "2020-04-19T07:22:00"],
  "recurring": "none"
}
```

Ensure your output is valid JSON. Only output the JSON, no extra text.

---

Figure 21: Prompt used for time range annotation over user questions.

---

Prompt for Fact–Evidence Alignment

---

Your task: Determine which utterance contains the most relevant evidence that supports each of the given facts.

**Input:**

- Facts: {`facts_list`}
- Sessions: {`sessions_data`}

**Output:** Return the most relevant utterance for each fact using the following format:

```
{
    "fact_evidence": [
        {
            "fact_index": 0,
            "session_id": "session_id",
            "utterance_id": "id"
        },
        {
            "fact_index": 1,
            "session_id": "session_id",
            "utterance_id": "id"
        },
        ...
    ]
}
```

**Example:**

```
{
    "fact_evidence": [
        {
            "fact_index": 0,
            "session_id": 2,
            "utterance_id": 3
        }
    ]
}
```

**Constraints:**

- Only include utterances that clearly support the fact (no hallucination or inference beyond what's stated).
- Select exactly ONE most relevant utterance per fact.
- If no utterance supports a fact, return `null` for that fact.
- `fact_index` corresponds to the index in the facts list (0-based).
- `session_id` must be one of the provided session IDs.

Figure 22: Prompt used for fact–evidence alignment in multi-session dialogues.

---

**Prompt for Memory-T1 Training**

You are a memory-aware reasoning assistant. Your task is to answer temporal questions based on multi-turn dialogue history. Carefully analyze the provided context, reason about time and events, and respond strictly in JSON format.
The required JSON structure is:
{ $"selected\_memory"$ : [$"session\_X"$, $"session\_Y"$], "answer": "X" }
Answer Format Rules (by type):
1. Single choice: A, B, ...
2. Multiple choice: A C E (space-separated)
3. Time: "HH:MM:SS am/pm, Month DD, YYYY"
Example: "02:30:00 pm, March 22, 2024"
4. Sequence: (1)(3)(2)(4)(5)(6)(8)(7)

Input format:

$< previous\_memory >${dialogue\_sessions}$< /previous\_memory >$
$< question >$
$Time : ${current\_time}$
$Question : ${question}$
$< /question >$
Output example:
{ $"selected\_memory"$ : [$"session\_1"$, $"session\_7"$], $"answer"$ : $"A"$ }

Figure 23: Prompt used for training Memory-T1.

---

**Prompt for Evaluation**

You are presented with a **temporal question** and a **previous memory**, please answer the question with the correct format. The last line of your response should be of the form: Answer: $Answer, where $Answer is the answer to the problem.
Output requirements:
1. Single choice: $A\|B\|$... (uppercase)
2. Multiple choice: A B C (space-separated uppercase)
3. Time: HH:MM:SS am/pm, Month DD, YYYY
Example: 10:45:41 pm, January 15, 2024 4. Sequence: (1)(3)(2)(4)(5)(6)(7)(8)

Input format:
$< previous\_memory >${dialogue\_sessions}$< /previous\_memory >$

$< question >$
$Time : ${current\_time}$
$Question : ${question}$
$< /question >$

Remember to put your answer on its own line after $'Answer'$:

Figure 24: Prompt used for evaluation of temporal reasoning tasks.

