# OpenReview forum: "Memory-T1: Reinforcement Learning for Temporal Reasoning in Multi-session Agents"
_ICLR.cc/2026/Conference — ICLR 2026 Poster_

### Official Review · Reviewer_AsHW · 2025-10-24

**Soundness:** 3
**Presentation:** 3
**Contribution:** 3
**Rating:** 8
**Confidence:** 4

**Summary:**

The paper proposes Memory-T1, a two-stage framework that improves the temporal reasoning ability of large language models (LLMs) over long, multi-session dialogues. During the first stage, an LLM predicts the temporal window and filters sessions outside the window. Then, a retriever reranks the resulting sessions, and only the top-ranked sessions are kept. During the second stage, an agent is trained with GRPO to select the most relevant sessions. The reward is a combination of accuracy, evidence grounding, and temporal consistency. The approach is applied to Qwen2.5-3B and Qwen2.5-7B-Instruct, and compared against diverse baselines.

**Strengths:**

- The paper addresses an important problem which is temporal reasoning on long, multi-session dialogues.
- The paper is clear and easy to read. The appendix contains additional details about the dataset and setup.
- Memory-T1 outperforms existing baselines, generalizes out-of-domain and is robust to long-context scenarios.
- The ablation study emphasizes the contribution of each reward.

**Weaknesses:**

- Unlike the base models, the performance when applying Memory-T1 does not seem to increase with model size (from 3B to 7B).
- Training requires a heavy annotation of the dataset which can be very expensive.
- There is no qualitative analysis showing the kind of mistakes of the base model that Memory-T1 solves.

**Questions:**

-  Memory-T1 does not seem to improve with model size. What could be the reason of this limitation? Is it more about the task than the approach?
- What is the additional latency introduced by the approach compared to the base model and the other baselines?
- Have you observed any reward hacking that you had to circumvent? Was it associated to particular reward or resulted from the combination of the rewards?
- Did you have to perform extensive hyperparameter search to obtain the results in the paper?
- Is it possible to fully automate dataset annotation?

---

> ### Author Response · Authors · 2025-11-24
> **Official Rebuttal to Weaknesses on Model Size and Qualitative Analysis**
>
> **No Improvement with Model Size**
>
> This limitation stems from the task’s nature, not the approach. Current LLMs (such as GPT-4) struggle with temporal reasoning, which does not scale proportionally with model size. Under this bottleneck, the 3B and 7B models exhibit similar baseline temporal understanding. After RL training with Memory-T1, the 3B model is already able to match the performance of the 7B model in temporal consistency and evidence alignment. In other words, the task’s temporal reasoning difficulty (not model capacity) limits scaling gains.
>
> **Qualitative Analysis of Model Mistakes**
>
> We analyzed subtasks where Memory-T1 shows the largest improvements (see table below). Across all six categories, a clear pattern emerges: the base model tends to retrieve evidence by semantic similarity rather than temporal correctness, leading to errors such as ignoring time windows, confusing event order, missing co-temporal alignments, and failing counterfactual adjustments. Memory-T1 addresses these issues through its time-range filtering and temporally consistent RL-based evidence selection, resulting in more accurate localization, ordering, and co-temporality. These behaviors directly explain the observed improvements on ER., OR., RR., CTF., Co-tmp., and Loc. tasks.
> | **Category** | **SubTask Name** | **Question** | **Options (A–D)** | **Base Model Answer (Wrong)** | **Memory-T1 Answer (Gold)** | **Base Model Overall Score** | **Memory-T1 Overall Score** |
> |-----------|--------|-----------------------|-------------------------------------|------|------|------|------|
> | **A. Loc.** | Localization | *When is Debra Ryan working on starting her own business?* | N/A | 9:32 pm, May 20, 2020 | **8:35 pm, February 21, 2020** | **0.278** | **0.500** |
> | **B. ER.** | Explicit Reasoning | *What notable artistic or outdoor activities did India Brown participate in between April 1, 2020, and April 9, 2020?* | A. India Brown attended a street art fest in Brazil.<br>B. India Brown took a photo of a feather and shells on a beach.<br>C. India Brown went hiking and sketching at a nearby national park.<br>D. India Brown received positive feedback on her artwork. | C | B | **0.692** | **0.821** |
> | **B. OR.** | Order Reasoning | *What was India Brown’s third teaching engagement in 2020?* | A. Running a painting workshop for kids.<br>B. Teaching art at an orphanage in Cambodia.<br>C. Conducting a live demonstration for her college art club.<br>D. Instructing a pottery class at a local studio. | **B** | **A** | **0.333** | **0.667** |
> | **B. RR.** | Relative Reasoning | *What was India Brown’s most recent job before 12:00 am, March 09, 2020?* | A. India Brown is working on a new series of abstract artworks based on her trip.<br>B. India Brown is working as a travel guide based on her trip experiences.<br>C. India Brown is working on a new painting technique learned at a street art festival.<br>D. India Brown is testing watercolors for her new series of abstract artworks. | B | A | **0.563** | **0.875** |
> | **C. CTF.** | Counterfactual | *What notable artistic or outdoor activities did India Brown participate in between April 1, 2020, and April 9, 2020, if she visited the Louvre in Paris in March 2020?* | A. India Brown carved a mini sculpture from a soap bar.<br>B. India Brown took a photo of a feather and shells on a beach.<br>C. India Brown took a photograph in Santorini, Greece.<br>D. India Brown sketched a waterfall during a hike. | C | B | **0.556** | **0.889** |
> | **C. Co-tmp.** | Co-temporality | *At the same time as Debra Ryan is learning to play the guitar, what collection does India Brown have?* | A. India Brown has a collection of soap sculptures.<br>B. India Brown has a collection of watercolor paintings.<br>C. India Brown has a collection of CDs.<br>D. India Brown has a collection of vinyl records. | **B** | **C** | **0.778** | **0.944** |

---

> > ### Author Response · Authors · 2025-11-24
> > **Official Rebuttal to Question**
> >
> > **Inference Latency**
> >
> > Memory-T1 incurs **negligible additional inference latency.** As shown below, its average latency **(1.26 seconds per query)** is comparable to baselines, and the retrieval overhead **(0.01 seconds)** is insignificant relative to LLM generation latency.
> >
> > | Model                 | Total Questions | Total Inference Time (s) | Avg Latency per Query (s) | Retrieval-aug Total Latency (s) | Retrieval Overhead per Query (s) |
> > | --------------------- | --------------- | ------------------------ | ------------------------- | ------------------------------- | -------------------------------- |
> > | **Memory-T1**         | 200             | 252.08                   | 1.26                      | 259.81                          | **0.01**                         |
> > | **Time-R1**           | 200             | 248.62                   | 1.24                      | 256.35                          | **0.01**                         |
> > | **MemAgent**          | 200             | 312.72                   | 1.56                      | 320.47                          | **0.01**                             |
> > | **Qwen2.5-3B (Base)** | 200             | 271.83                   | 1.36                      | 279.74                          | **0.01**                         |
> >
> > **Reward Hacking**
> >
> > No reward hacking was observed during training. This is mainly because the temporal reward is deliberately designed to be weak and bounded, and it cannot be optimized independently of the accuracy and grounding rewards. The three rewards reinforce each other and penalize inconsistent behavior, which makes it difficult for the model to exploit any single reward signal. As a result, no special mechanisms for preventing reward hacking were needed.
> >
> > **Hyperparameter Search**
> >
> > Strong initial performance was achieved without tuning. A subsequent grid search over weight combinations (a sanity check, not extensive tuning) confirmed the method’s stability across settings.
> >
> > **Fully Automated Dataset Annotation**
> >
> > Yes. After each round of model-generated annotations, we can employ a stronger LLM to perform reflection and error attribution, effectively replacing human validators. This automatic error-attribution stage achieves performance comparable to human annotators. With this closed-loop data flywheel, the dataset annotation process can be fully automated.
> >
> > Many thanks for your valuable comments.

---

### Official Review · Reviewer_rTBM · 2025-10-27

**Soundness:** 3
**Presentation:** 3
**Contribution:** 2
**Rating:** 4
**Confidence:** 3

**Summary:**

This paper presents Memory-T1, a reinforcement learning framework designed to improve temporal reasoning capabilities in multi-session conversational agents. The authors address a critical challenge: existing long-context models struggle to accurately identify temporally relevant information from noisy, extended dialogue histories. The framework employs a two-phase architecture: 1) Candidate Generation: LLM first predicts its target temporal window followed by relevance-based retrieval (BM25) to narrow the search space; 2) Fine-grained Selection: Employs an RL agent trained with GRPO to select precise evidence sessions. A multi-level reward function is designed to jointly optimize the evidence selection and answer prediction. The framework achieves 67.0% overall score on the Time-Dialog benchmark with Qwen-2.5-7B, outperforming a Qwen-2.5-14B baseline by 10.2%.

**Strengths:**

- The coarse-to-fine retrieval strategy combined with multi-level RL rewards is well-justified. The temporal consistency reward design enhanced the temporal reasoning and evidence selection of the model to better predict the answer.
- Ablation studies (Table 2) provide clear evidence for each reward component's contribution, demonstrating the importance of individual design choices.

**Weaknesses:**

- The primary evaluation is conducted on the in-domain Time-Dialog dataset, where Memory-T1 is trained in-domain while other baselines (Time-R1, MemAgent) are evaluated in a zero-shot setting. This makes direct comparison difficult. It remains unclear how Memory-T1 would perform against these baselines on unseen benchmarks such as LoCoMo, particularly in a comparable experimental setup.
- The proposed framework requires extensive temporal annotations for training, yet the sensitivity to annotation quality and potential noise has not been explored. This raises concerns about the framework's practical applicability.

**Questions:**

What happens when the LLM-based temporal filter incorrectly predicts the query time scope and excludes ground-truth evidence sessions? Does the framework include any mechanism to allow error recovery from such retrieval failures?

---

> ### Author Response · Authors · 2025-11-24
> **Official Rebuttal to Weaknesses**
>
> **Out-of-Domain Generalization**
>
> To address the in-domain bias, we evaluated Time-R1, MemAgent and Memory-T1 on the unseen LoCoMo benchmark under the same RAG setting. The performance comparison is summarized below:
>
> | Model Family   | Params | Setting | F1 Score | Setting | F1 Score |
> | -------------- | -------|---------|------------------|------------------|------------------|
> | Time-R1        | 3B     | RAG     | 31.4             | Non-RAG | 29.2         |
> | MemAgent       | 7B     | RAG     | 37.6             | Non-RAG | 40.2         |
> | **Memory-T1**  | 3B     | RAG     | 36.7         | Non-RAG | 37.7     |
>
> Memory-T1 (3B) matches MemAgent (7B) and significantly outperforms  Time-R1 (3B), demonstrating strong out-of-domain generalization despite a smaller backbone.
>
> **Sensitivity to Annotation Noise**
>
> We conducted a sensitivity analysis by explicitly corrupting training time labels (5%, 10%, 20 noise) with perturbations (±7/30/180/360 days, mimicking real annotation errors). Results are shown below:
>
> |      | Loc. (A) | DC. (A) | Comp. (A) | OC. (A) | Ext. (A) | ER. (B) | OR. (B) | RR. (B) | CTF. (C) | Co-tmp. (C) | TL. (C) | Overall |
> | ---- | -------- | ------- | --------- | ------- | -------- | ------- | ------- | ------- | -------- | ----------- | ------- | ------- |
> | 20%  | 27.8     | 43.5    | 5.0       | 55.0    | 25.9     | 76.9    | 72.2    | 81.2    | 94.4     | 94.4        | 16.7    | 60.0    |
> | 10%  | 50.0     | 43.5    | 10.6      | 65.0    | 55.6     | 74.4    | 67.4    | 81.2    | 88.9     | 94.4        | 18.8    | 63.4    |
> | 5%   | 50.0     | 60.9    | 5.0       | 60.0    | 55.6     | 82.0    | 77.8    | 87.5    | 94.4     | 88.9        | 16.7    | 67.0    |
>
> With 5% noise (realistic error rate), overall F1 remains 67.0, and key temporal reasoning tasks such as Counterfactual (CTF.), Co-temporality (Co-tmp.), and Relative Reasoning (RR.) stay high at 94.4, 88.9, and 87.5, respectively. Increasing the noise to 10% and 20% leads to a gradual but moderate degradation of the overall score to 63.4 and 60.0. Notably, the most temporally demanding tasks remain robust: CTF. and Co-tmp. stay above 88.9 F1 even at 20% noise. The main decline is concentrated in time-span–related subtasks (such as Localization and Extract). This confirms the framework is resilient to realistic label noise, supporting its practical applicability in real-world settings where time labels are imperfect.
>
> **Recovery from Temporal Filter Errors**
>
> Thank you for the insightful question. The temporal filter uses a **conservative pre-filtering strategy**:  predicted ranges are widened with a safety margin to include not only the “exact” time spans but also neighboring sessions. As shown in **Figure 3**, temporal filtering achieves **higher ground-truth recall than vanilla RAG across all K values, reducing retrieval noise and improving access to correct evidence.**  While no explicit fallback mechanism exists yet, this conservative design minimizes ground-truth exclusion risks. Adding fallback (e.g., re-retrieving without temporal filters) is a promising future extension.

---

### Official Review · Reviewer_mMws · 2025-11-01

**Soundness:** 3
**Presentation:** 4
**Contribution:** 2
**Rating:** 6
**Confidence:** 3

**Summary:**

This paper presents Memory-T1, a reinforcement learning framework designed to improve temporal reasoning in long, multi-session dialogue settings. The method introduces a two-stage process: a coarse retrieval stage that filters relevant sessions by predicting temporal scope and ranking past interactions, followed by a fine retrieval stage where a reinforcement learning agent selects evidence guided by multi-level rewards. The model uses three reward components related to answer accuracy, evidence grounding, and temporal consistency, aiming to enhance both factual correctness and chronological alignment. Experiments on Time-Dialog and LoCoMo show that Memory-T1 improves temporal reasoning accuracy and robustness under long-context conditions.

**Strengths:**

The paper targets an important and underexplored aspect of dialogue modeling, namely temporal reasoning across multiple sessions. The proposed coarse-to-fine retrieval framework is intuitive and improves retrieval efficiency under long histories. Integrating reinforcement learning provides a structured way to optimize multiple supervision signals jointly. The experimental results demonstrate strong improvements on established benchmarks, and the ablation studies highlight the contribution of each reward type. The paper is clearly written, and the motivation is easy to follow.

**Weaknesses:**

The methodological novelty is moderate. The framework largely combines existing reinforcement learning and retrieval techniques with additional temporal supervision. The temporal consistency reward is well-motivated but not particularly new, and its formulation appears heuristic. The paper does not provide enough justification for the chosen reward weights or a systematic analysis of their sensitivity. The ablation results are limited and do not clearly establish whether improvements stem from the reward design itself or from better retrieval heuristics. The method also assumes access to accurate timestamps and event annotations, which may limit applicability to real-world data where such information is noisy or missing. Finally, the evaluation focuses on QA-style reasoning without exploring whether the learned policy improves temporal coherence in free-form dialogue generation.

**Questions:**

- How sensitive is the model to the relative weights of the three reward components?
- Were the weights tuned manually or derived from validation metrics?
- Could the method work in domains without explicit timestamps or structured temporal annotations?
- How much of the observed improvement comes from the reward design itself versus better retrieval filtering?
- Would the learned policy transfer to different backbone models or datasets?

---

> ### Author Response · Authors · 2025-11-24
> **Official Rebuttal to Weaknesses**
>
> **Weakness & Question Response**
>
> Our core novelty lies not only in designing new RL algorithms, but also in introducing temporal supervision into multi-session memory selection, which is an under explored yet critical direction for long-horizon agents. While building on existing RL and retrieval components, we **explicitly optimize cross-session temporal consistency** in memory-selection, a gap unaddressed by static retrieval/filtering methods (which lack temporal reasoning mechanisms). Notably, **temporal annotations are only used for RL reward construction, not inference**, where the model only takes raw dialogue and session-level timestamps (fully aligned with real-world data). This enables the policy to learn timeline-aware behaviors (e.g., cross-session evidence selection) that static heuristics cannot capture.
>
> **Reward-weight Sensitivity**
>
> We appreciate the concern regarding the choice of reward weights and conducted a small-scale sensitivity analysis (Appendix Figure 6) with four weight configurations. The model consistently outperformed baselines across all settings, leading us to select the final weights of **0.6 (accuracy)**, **0.2 (evidence grounding)**, and **0.2 (time consistency)**. This choice was informed by empirical validation rather than manual intuition alone.
>
> **Source of Performance Gains**
>
> Controlled experiments isolate RL’s impact: **Figure 3** shows Memory-T1 (blue bars) outperforms Qwen2.5-3B (green bars) with identical RAG pipelines, proving gains stem from the RL policy, not retrieval heuristics. **Table 2’s** ablation study (removing individual reward) demonstrates performance degradation, confirming Memory-T1’s improvement emerges from the **effective combination** of the three reward terms, rather than from any single heuristic.
>
> **Applicability to Noisy Annotations.**
>
> **Event timestamps / annotations are only used for RL reward computation, not inference.** At deployment, the model requires only **the dialogue content** and **the session-level timestamps** (readily available in real-world logs), avoiding reliance on fine-grained utterance-level annotations, ensuring practical applicability.
>
> **QA-style evaluation.**
>
> We acknowledge the value of free-form dialogue evaluation. However, most temporal-reasoning and long-term-memory benchmarks (e.g., TIME, LoCoMo) adopt **QA-style evaluation** for objective accuracy-based comparison (free-form generation depends heavily on subjective LLM-based evaluators). To maintain comparability, we followed this standard and explicitly list open-end dialogue evaluation as future work.
>
> Many thanks for your valuable comments.

---

### Official Review · Reviewer_XvHc · 2025-11-04

**Soundness:** 3
**Presentation:** 4
**Contribution:** 4
**Rating:** 4
**Confidence:** 3

**Summary:**

This paper proposes Memory-T1, a two-stage framework for temporal reasoning over long, multi-session dialogues. Stage-1 performs coarse-to-fine candidate generation: an LLM predicts the query’s temporal window to hard-filter sessions, then BM25 ranks by textual relevance to form a high-recall pool. Stage-2 fine-tunes a policy with GRPO to jointly pick evidence sessions and produce the answer, supervised by a multi-level reward that combines task accuracy (Ra), evidence grounding via session-set overlap (Rg), and a temporal-consistency reward (Rt) with session-level proximity (Rs) and utterance-level temporal fidelity (Rf).

On Time-Dialog, Memory-T1 improves a 3B model to 66.9% overall, and the 7B variant reaches 67.0%, outperforming larger baselines. The ablations attribute gains to the temporal-consistency and grounding rewards; removing them substantially degrades performance. The method shows strong OOD results on LoCoMo (37.7% overall without RAG), and robustness up to 128k tokens.

**Strengths:**

1. Timely and important: Tackles temporal reasoning over long multi-session dialogues—an increasingly central capability for agentic systems and memory architectures.

2. Well-motivated reward design for GRPO: Clear decomposition into Ra/Rg/Rt; Rt thoughtfully mixes proximity and fidelity with soft penalties and explicit hyper-parameters; weights and sensitivity are reported.

3. Coarse-to-fine retrieval is executed cleanly with a precise time filter then lexical ranking; the analysis of top-k and recall is helpful.

4. Clear writing and comprehensive experiments: SOTA on Time-Dialog, detailed ablations of reward components, OOD evaluation (LoCoMo), and long-context stress tests.

**Weaknesses:**

(W1) Table 2 interpretation of Rt/Rs/Rf effects

The paper states that removing Rs or Rf yields a trade-off aligned with task difficulty (Category-A “simpler” tasks improve, while B/C “complex” tasks degrade). However the full removal of Rt (both Rs and Rf) does not show a monotone extension of the same trend.
If Category-A improvements under −Rs/−Rf are due to “easier” temporal structure, why does removing the entire Rt lower Category-A below the full model instead of amplifying that benefit? Two possibilities worth probing:
1. Reward-interaction / reweighting hypothesis. With wt=0, the optimizer may over-fit Ra+Rg. E.g. Ra for Category-A can reward surface-level answers
2. Credit-assignment : GRPO with a batch-average baseline can shift exploration when a dense component disappears. Without Rt, advantages may over-credit generations that incidentally match Ra/Rg


(W2) Table 3 (LoCoMo) and how RAG is integrated with Memory-T1

Table 3 shows Memory-T1 is best without RAG overall (37.7% vs. 36.7%), supporting the claim that the policy “manages memory” well. But on the Adversarial subset, RAG helps Memory-T1 quite noticeably (29.8 with RAG vs. 26.0 without). Why? Can the authors explain this part?
How, precisely, is RAG layered over Memory-T1’s time filter + BM25 + RL policy (i.e., is RAG material merged before/after candidate generation, and does it change the action space for evidence selection)? Please detail.

One possibility here: distribution-shift hypothesis -- the RL policy learns a specialized memory manager for in-dialogue, time-anchored evidence. Injecting external RAG text can introduce off-distribution segments with weak or missing temporal anchors, which lowers precision overall.

**Questions:**

See weakness session

---

> ### Author Response · Authors · 2025-11-24
> **Official Rebuttal to [W1]**
>
> **Table 2 interpretation of Rt/Rs/Rf effects**
>
> Thank the reviewer for the insightful observation. The seeming non-monotonic behavior under full removal of **Rt** arises from the temporal distance computation task in **Category-A** tasks (e.g., computing days between two implicitly referenced events). These tasks rely on two distinct temporal consistency mechanisms:
> - **Session-level temporal consistency** (captured by Rs), which constrains the global timeline across sessions.
> - **Content-level temporal relevance** (captured by Rf), which guides the model select dialogue segments whose temporal clues are relevant to the queried event pair.
>
> Category A focuses more on a single dominant temporal signal (like identifying timestamp and event duration computation). When removing Rs or Rf individually, the remaining component still provides a compensatory constraint: removing Rs simplifies session-level timeline, but Rf preserves relevant content selection (benefiting timestamp extraction), while removing Rf maintains timeline reasoning (aiding event ordering or gap measurement). When Rf is added on top of Rs, the two rewards may produce partially misaligned gradients, that is why Rs+Rf is lower than Rs-only and Rf-only.
>
> However,**removing Rt eliminates the system’s ability to jointly consider (i) correct event selection and (ii) consistent temporal alignment.** For duration-based Category-A tasks, the model may misidentify the relevant events or rely solely on conversational order, resulting in incorrect length estimation and consequently **lower performance than the full Memory-T1.**
>
> In short, the improvement under −Rs/−Rf stems from reduced temporal rigidity with a remaining compensatory mechanism, whereas full Rt removal eliminates both regulating factors and leads to compounded errors. Hence, the observed pattern is consistent with the underlying task structure.

---

> > ### Author Response · Authors · 2025-11-24
> > **Official Rebuttal to [W2]**
> >
> > **Memory-T1-RAG interaction**
> >
> >  We thank the reviewer for highlighting this and agree the Memory-T1-RAG interaction deserves clearer explanation. In all settings:
> >
> > 1.	We first conduct retrieval via our time-aware memory module: for a given query, Memory-T1 applies a time filter followed by a relevance filter to the multi-session dialogue history, yielding a (potentially long) candidate pool of sessions/utterances.
> > 2.	The **RL-trained policy** then takes this candidate pool as input and **selects evidence sessions while generating the answer**, learning to maintain temporal consistency over long histories.
> > Memory-T1 only receives (i) raw multi-session dialogue and (ii) session-level timestamps (naturally available in real-world scenarios) at inference; fine-grained time anchors exclusively used in the reward function for RL training, not as model inputs. This specialization enables the policy to select in-dialogue, time-anchored evidence via reward shaping, not inference time annotation.
> >
> > RAG introduces an inherent precision–recall trade-off:
> >
> > - Small top-k → high precision, good for answerability, but may drop key temporal evidence.
> > - Large top-k → high recall and better support for timeline reasoning but introduces noise and mild distribution shift.
> > Memory-T1 navigates this trade-off by using RAG only as a light pruning step while retaining a large enough top-k to preserve the temporal structure needed for the RL policy.
> >
> > **Why overall accuracy slightly drop, but the Adversarial subset improves?**
> >
> > The **Adversarial subtask** focus on **answerability detection** (saying “I don’t know” when the information is missing):
> > - Without RAG: Post-filter candidate pool remains lengthy, prone to “lost in the middle” effects and spurious snippets that encourage hallucination.
> > - With RAG: the **condensed candidate pool prunes** spurious in-dialogue segments, preserving a compact set lacking supporting evidence. This makes it **easier for the RL policy to learn “unanswerable” behavior** more effectively (26.0 → 29.8).
> >
> > In summary, RAG acts as a **pre-generation retrieval augmentation** that modifies the candidate evidence distribution (not policy structure). It introduces mild distribution shift and loss of temporally key candidates on standard tasks but benefits adversarial detection by simplifying evidence incompleteness detection. We will clarify this integration and its implications in the revised manuscript.

---

### Author Response · Authors · 2025-12-01
**Summary of Rebuttal to Reviewer Comments**

We sincerely thank all the reviewers for their insightful and constructive comments, which have greatly contributed to improving our work. We have thoroughly revised the paper to address the reviewers' concerns. Below we summarize the major revisions (the main revisions are marked with blue text in the pdf), while we reply to the comments of each reviewer separately.

**To Reviewer XvHc**

- **W1 Weakness On Reward Ablation (W1):** Clarified that the observed trade-offs when removing specific reward components ($R_s$ or $R_f$) arise from the compensatory nature of the reward design.  We have incorporated the new explanation in the rebuttal response and made necessary revisions to the manuscript (**lines 427-431**).
- **W2 On RAG Integration & Adversarial Results (W2):** We provided Memory-T1-RAG interaction in the official response and added analysis on Adversarial Subtask Results from LoCoMo to the manuscript (**lines 464-475**).

**To Reviewer mMws**

- **W1 On Novelty:** Argued that the core contribution is not only in designing new RL algorithms, but also in introducing temporal supervision into multi-session memory selection, which is an underexplored yet critical direction for long-horizon agents. More details are shown in Rebuttal Response.
- **W1 QA-style evaluation.** We acknowledge the value of free-form dialogue evaluation. We followed the standard QA-style evaluation similar to benchmarks (e.g., TIME, LoCoMo).
- **Q1,2 On Reward Sensitivity:**  We clarified that our sensitivity analysis is shown in **Appendix Fig 6**, showing the model is stable across weight configurations.
- **Q3 On Reliance on Annotations:** Clarified that fine-grained annotations are **only for training**. Inference requires only raw text and session-level logs (standard in real-world scenarios).
- **Q4 On Source of Gains (RL vs. Retrieval):** Pointed to controlled experiments (**Fig 3 and Table 2**) where Memory-T1 outperforms baselines using *identical* retrieval pipelines, isolating the gains to the RL policy.
- **Q5 Transfer to different backbone models or datasets.** Provided new comparison results on the **unseen LoCoMo benchmark**. Memory-T1 (3B) matches the larger MemAgent (7B) and significantly outperforms Time-R1 (3B), proving strong generalization. The results are shown in revised manuscript (**Table 3, Appendix Table 9**)

**To Reviewer rTBM**

- **W1 On In-Domain Bias & OOD:** Provided new comparison results on the **unseen LoCoMo benchmark**. Memory-T1 (3B) matches the larger MemAgent (7B) and significantly outperforms Time-R1 (3B), proving strong generalization. The results are shown in revised manuscript (**Appendix Table 9**)
- **W2 On Annotation Noise Sensitivity:** Conducted a new experiment adding 5-20% noise to training labels. The model remains robust (67.0 $\to$ 60.0 F1). The details are shown in revised manuscript (**Table 4, lines 504-512**)
- **Q1 On Retrieval Failure Recovery:** Explained the "conservative pre-filtering" strategy (widened search windows), which achieves higher recall than vanilla RAG, minimizing exclusion risks.

**To Reviewer AsHW**

- **W1,Q1 On Scaling Limitations (3B vs 7B):** Attributed the lack of scaling to the inherent difficulty of temporal reasoning tasks for current LLMs (the task is the bottleneck). Highlighted that Memory-T1 allows a 3B model to match 7B performance.
- **W3 On Qualitative Analysis:** Provided a detailed table in the revised manuscript (**line 520-527, Appendix Table 10**) of examples showing Memory-T1 correcting base model errors (e.g., ignoring time windows, confusing event order) via temporal consistency.
- **Q2 On Inference Latency:** Presented data showing negligible overhead (0.01s retrieval vs. ~1.26s total latency), confirming efficiency, the details can be found in revised manuscript (**lines 514-518, Table 5**)
- **Q3 Reward Hacking**: Explained that no reward hacking was observed.
- **Q4 Hyperparameter Search**: Clarified that strong performance was achieved without extensive tuning. A subsequent grid search served as a sanity check to confirm stability across settings.(**Table 6**)
-  **W2, Q5 Automated Dataset Annotation**: Confirmed that the process can be fully automated using a closed-loop data flywheel, where a stronger LLM performs reflection and error attribution, achieving performance comparable to human validators.

---

### Meta-Review · Area_Chair_Dy7Q · 2026-01-07

**Summary:**

The paper proposes Memory-T1, a reinforcement learning (RL)–based framework for temporal reasoning over long, multi-session dialogues. Memory-T1 adopts a coarse-to-fine, multi-stage, first filtering relevant sessions by predicting temporal scope and ranking past interactions, followed by an RL-trained policy that selects precise evidence while generating answers. Training is guided by a multi-component reward that combines task accuracy, evidence grounding, and temporal consistency at both session and utterance levels. Experimental results on Time-Dialog and LoCoMo demonstrate improved temporal reasoning accuracy, robustness to long contexts, and strong performance relative to larger baselines, including cases where a smaller backbone matches or exceeds larger models.

The paper was evaluated by four referees who agree on many of the paper's key strengths and weaknesses. As most if not all reviewers emphasize, the paper addresses an important and timely problem, namely temporal reasoning over extended multi-session dialogues. Additionally, the reviewers appreciate the execution of the proposed coarse-to-fine strategy, they emphasize the performance gains on established benchmarks the ablated benefits of the contributions of each reward type, and find that the paper is clear and well-written. At the same time, the reviewers share several concerns about the paper as initially submitted. These include questions of methodological novelty in what reviewers characterize as the integration of existing RL and retrieval components with additional temporal supervision, which at least one reviewer finds to be heuristic in nature. At least three reviewers are concerned about the amount of accurate temporal supervision that is required. Other issues raised by the reviewers include a limited analysis of reward sensitivity, questions about the fairness of the experimental evaluation, and the lack of clear scaling gains as model size increases.

The authors made a concerted effort to address the reviewers' questions and concerns. This included the addition of new experiments that demonstrate the method's robustness to some amount of noise in the temporal annotations, experiments on a new benchmark that does not exhibit potential in-domain bias,  clarifications regarding reward sensitivity, and an evaluation that shows the contributions of RL compared to retrieval, among others. Overall, the rebuttal helps to clarify the empirical benefits of the proposed framework, key design choices, and the extent of the paper's overall contributions.

**Reviewer Concerns:**

The authors made a concerted effort to address the reviewers' questions and concerns. This included the addition of new experiments that demonstrate the method's robustness to some amount of noise in the temporal annotations, experiments on a new benchmark that does not exhibit potential in-domain bias,  clarifications regarding reward sensitivity, and an evaluation that shows the contributions of RL compared to retrieval, among others. Overall, the rebuttal helps to clarify the empirical benefits of the proposed framework, key design choices, and the extent of the paper's overall contributions.

**Reviewer Scores:**

Several of the reviewers' key concerns were addressed by the rebuttal.

---

### Decision · Program_Chairs · 2026-01-26

Accept (Poster)